# The Convex Geometry of Backpropagation: Neural Network Gradient Flows Converge to Extreme Points of the Dual Convex Program

**Yifei Wang**
Department of Electrical Engineering
Stanford University
Stanford, CA 94305, USA
wangyf18@stanford.edu

**Mert Pilanci**
Department of Electrical Engineering
Stanford University
Stanford, CA 94305, USA
pilanci@stanford.edu

## Abstract

We study non-convex subgradient flows for training two-layer ReLU neural networks from a convex geometry and duality perspective. We characterize the implicit bias of unregularized non-convex gradient flow as convex regularization of an equivalent convex model. We then show that the limit points of non-convex subgradient flows can be identified via primal-dual correspondence in this convex optimization problem. Moreover, we derive a sufficient condition on the dual variables which ensures that the stationary points of the non-convex objective are the KKT points of the convex objective, thus proving convergence of non-convex gradient flows to the global optimum. For a class of regular training data distributions such as orthogonal separable data, we show that this sufficient condition holds. Therefore, non-convex gradient flows converge to optimal solutions of a convex optimization problem. We present numerical results verifying the predictions of our theory for non-convex subgradient descent.

## 1 Introduction

Neural networks (NNs) exhibit remarkable empirical performance in various machine learning tasks. However, a full characterization of the optimization and generalization properties of NNs is far from complete. Non-linear operations inherent to the structure of NNs, over-parameterization and the associated highly nonconvex training problem makes their theoretical analysis quite challenging.

In over-parameterized models such as NNs, one natural question arises: Which particular solution does gradient descent/gradient flow find in unregularized NN training problems? Suppose that $\mathbf{X} \in \mathbb{R}^{N \times d}$ is the training data matrix and $\mathbf{y} \in \{1, -1\}^N$ is the label vector. For linear classification problems such as logistic regression, it is known that gradient descent (GD) exhibits implicit regularization properties, see, e.g., (Soudry et al., 2018; Gunasekar et al., 2018). To be precise, under certain assumptions, GD converges to the following solution which maximizes the margin:

$$\arg\min_{\mathbf{w} \in \mathbb{R}^d} \frac{1}{2}\|\mathbf{w}\|_2^2, \text{ s.t. } y_n \mathbf{w}^T \mathbf{x}_n \geq 1, n \in [N]. \tag{1}$$

Here we denote $[N] = \{1, \ldots, N\}$. Recently, there are several results on the implicit regularization of the (stochastic) gradient descent method for NNs. In (Lyu & Li, 2019), for the multi-layer homogeneous network with exponential or cross-entropy loss, with separable training data, it is shown that the gradient flow (GF) and GD finds a stationary point of the following non-convex max-margin problem:

$$\arg\min_{\boldsymbol{\theta}} \frac{1}{2}\|\boldsymbol{\theta}\|_2^2, \text{ s.t. } y_n f(\boldsymbol{\theta}; \mathbf{x}_n) \geq 1, n \in [N], \tag{2}$$

where $f(\boldsymbol{\theta}; \mathbf{x})$ represents the output of the neural network with parameter $\boldsymbol{\theta}$ given input $\mathbf{x}$. In (Phuong & Lampert, 2021), by further assuming the orthogonal separability of the training data, it is shown that all neurons converge to one of the two max-margin classifiers. One corresponds to the data with positive labels, while the other corresponds to the data with negative labels.

However, as the max-margin problem of the neural network (2) is a non-convex optimization problem, the existing results only guarantee that it is a stationary point of (2), which can be a local minimizer or even a saddle point. In other words, the global optimality is not guaranteed.

In a different line of work (Pilanci & Ergen, 2020; Ergen & Pilanci, 2020; 2021b), exact convex optimization formulations of two and three-layer ReLU NNs are developed, which have global optimality guarantees in polynomial-time when the data has a polynomial number of hyperplane arrangements, e.g., in any fixed dimension or with convolutional networks of fixed filter size. The convex optimization framework was extended to vector output networks (Sahiner et al., 2021b), quantized networks (Bartan & Pilanci, 2021b), autoencoders (Sahiner et al., 2021c; Gupta et al., 2021), networks with polynomial activation functions (Bartan & Pilanci, 2021a), networks with batch normalization (Ergen et al., 2021), univariate deep ReLU networks, deep linear networks (Ergen & Pilanci, 2021c) and Generative Adversarial Networks (Sahiner et al., 2021a).

In this work, we first derive an equivalent convex program corresponding to the maximal margin problem (2). We then consider non-convex subgradient flow for unregularized logistic loss. We show that the limit points of non-convex subgradient flow can be identified via primal-dual correspondence in the convex optimization problem. We then present a sufficient condition on the dual variable to ensure that all stationary points of the non-convex max-margin problem are KKT points of the convex max-margin problem. For certain regular datasets including orthogonal separable data, we show that this sufficient condition on the dual variable holds, thus implies the convergence of gradient flow on the unregularized problem to the global optimum of the non-convex maximalo margin problem (2). Consequently, this enables us to fully characterize the implicit regularization of unregularized gradient flow or gradient descent as convex regularization applied to a convex model.

## 1.1 RELATED WORK

There are several works studying the property of two-layer ReLU networks trained by gradient descent/gradient flow dynamics. The following papers study the gradient descent like dynamics in training two-layer ReLU networks for regression problems. Ma et al. (2020) show that for two-layer ReLU networks, only a group of a few activated neurons dominate the dynamics of gradient descent. In (Mei et al., 2018), the limiting dynamics of stochastic gradient descent (SGD) is captured by the distributional dynamics from a mean-field perspective and they utlize this to prove a general convergence result for noisy SGD. Li et al. (2020) focus on the case where the weights of the second layer are non-negative and they show that the over-parameterized neural network can learn the ground-truth network in polynomial time with polynomial samples. In (Zhou et al., 2021), it is shown that mildly over-parameterized student network can learn the teacher network and all student neurons converge to one of the teacher neurons.

Beyond (Lyu & Li, 2019) and (Phuong & Lampert, 2021), the following papers study the classification problems. In (Chizat & Bach, 2018), under certain assumptions on the training problem, with over-parameterized model, the gradient flow can converge to the global optimum of the training problem. For linear separable data, utilizing the hinge loss for classification, Wang et al. (2019) introduce a perturbed stochastic gradient method and show that it can attain the global optimum of the training problem. Similarly, for linear separable data, Yang et al. (2021) introduce a modified loss based on the hinge loss to enable (stochastic) gradient descent find the global minimum of the training problem, which is also globally optimal for the training problem with the hinge loss.

## 1.2 PROBLEM SETTING

We focus on two-layer neural networks with ReLU activation, i.e., $f(\boldsymbol{\theta}, \mathbf{X}) = (\mathbf{X}\mathbf{W}_1)_+\mathbf{w}_2$, where $\mathbf{W}_1 \in \mathbb{R}^{d \times m}$, $\mathbf{w}_2 \in \mathbb{R}^m$ and $\boldsymbol{\theta} = (\mathbf{W}_1, \mathbf{w}_2)$ represents the parameter. Due to the ReLU activation, this neural network is homogeneous, i.e., for any scalar $c > 0$, we have $f(c\boldsymbol{\theta}; \mathbf{X}) = c^2 f(\boldsymbol{\theta}; \mathbf{X})$. The training problem is given by

$$\min_{\boldsymbol{\theta}} \sum_{n=1}^{N} \ell(y_n f(\boldsymbol{\theta}; \mathbf{x}_n)), \tag{3}$$

where $\ell(q) : \mathbb{R} \to \mathbb{R}_+$ is the loss function. We focus on the logistic, i.e, cross-entropy loss, i.e., $\ell(q) = \log(1 + \exp(-q))$.

Then, we briefly review gradient descent and gradient flow. The gradient descent takes the update rule

$$\boldsymbol{\theta}(t+1) = \boldsymbol{\theta}(t) - \eta(t)\mathbf{g}(t),$$

where $\mathbf{g}(t) \in \partial^\circ \mathcal{L}(\boldsymbol{\theta}(t))$ and $\partial^\circ$ represents the Clarke's subdifferential.

The gradient flow can be viewed as the gradient descent with infinitesimal step size. The trajectory of the parameter $\boldsymbol{\theta}$ during training is an arc $\boldsymbol{\theta} : [0, +\infty) \to \Theta$, where $\Theta = \{\boldsymbol{\theta} = (\mathbf{W}_1, \mathbf{w}_2) | \mathbf{W}_1 \in \mathbb{R}^{d \times m}, \mathbf{W}_2 \in \mathbb{R}^m\}$. More precisely, the gradient flow is given by the differential inclusion

$$\frac{d}{dt}\boldsymbol{\theta}(t) \in -\partial^\circ \mathcal{L}(\boldsymbol{\theta}(t)), \quad \text{for } t \geq 0, \text{ almost everywhere.}$$

## 2 MAIN RESULTS

In this section, we present our main results and defer the detailed analysis to the following sections. Consider the more general multi-class version of the problem with $K$ classes. Suppose that $\bar{\mathbf{y}} \in [K]^N$ is the label vector. Let $\mathbf{Y} = (y_{n,k})_{n \in [N], k \in [K]} \in \mathbb{R}^{N \times K}$ be the encoded label matrix such that

$$y_{n,k} = \begin{cases} 1, & \text{if } \bar{y}_n = k, \\ -1, & \text{otherwise.} \end{cases}$$

Similarly, we consider the following two-layer vector-output neural networks with ReLU activation:

$$F(\boldsymbol{\Theta}, \mathbf{X}) = \begin{bmatrix} f_1(\boldsymbol{\theta}_1, \mathbf{X}) \\ \vdots \\ f_K(\boldsymbol{\theta}_K, \mathbf{X}) \end{bmatrix} = \begin{bmatrix} (\mathbf{X}\mathbf{W}_1^{(1)})_+ \mathbf{w}_2^{(1)} \\ \vdots \\ (\mathbf{X}\mathbf{W}_1^{(K)})_+ \mathbf{w}_2^{(K)} \end{bmatrix},$$

where we write $\boldsymbol{\Theta} = (\boldsymbol{\theta}_1, \ldots, \boldsymbol{\theta}_K)$. For $k = 1, \ldots, K$, we have $\boldsymbol{\theta}_k = (\mathbf{W}_1^{(k)}, \mathbf{w}_2^{(k)})$ where $\mathbf{W}_1^{(k)} \in \mathbb{R}^{N \times m}$ and $\mathbf{w}_2^{(k)} \in \mathbb{R}^m$. One can view each of the $K$ outputs of $F(\boldsymbol{\Theta}, \mathbf{X})$ as the output of a two-layer scalar-output neural network. Consider the following training problem:

$$\min_{\boldsymbol{\Theta}} \sum_{k=1}^K \sum_{n=1}^N \ell(y_{n,k} f_k(\boldsymbol{\theta}_k, \mathbf{x}_n)). \tag{4}$$

According to (Lyu & Li, 2019), the gradient flow and the gradient descent finds a stationary point of the following non-convex max-margin problem:

$$\arg\min_{\boldsymbol{\Theta}} \sum_{k=1}^K \frac{1}{2} \|\boldsymbol{\theta}_k\|_2^2, \text{ s.t. } y_{n,k} f(\boldsymbol{\theta}_k; \mathbf{x}_n) \geq 1, n \in [N], k \in [K]. \tag{5}$$

Denote the set of all possible hyperplane arrangement as

$$\mathcal{P} = \{\mathbf{diag}(\mathbb{I}(\mathbf{X}\mathbf{w} \geq 0)) | \mathbf{w} \in \mathbb{R}^d\}, \tag{6}$$

and let $p = |\mathcal{P}|$. We can also write $\mathcal{P} = \{\mathbf{D}_1, \ldots, \mathbf{D}_p\}$. From (Cover, 1965), we have an upper bound $p \leq 2r \left(\frac{e(N-1)}{r}\right)^r$ where $r = \text{rank}(X)$. We first reformulate (5) as convex optimization.

**Proposition 1** *The non-convex problem* (5) *is equivalent to the following convex program*

$$\min \sum_{k=1}^K \sum_{j=1}^p (\|\mathbf{u}_{j,k}\|_2 + \|\mathbf{u}'_{j,k}\|_2),$$

$$\text{s.t. } \mathbf{diag}(\mathbf{y}_k) \sum_{j=1}^p \mathbf{D}_j \mathbf{X}(\mathbf{u}_{j,k} - \mathbf{u}'_{j,k}) \geq \mathbf{1}, \tag{7}$$

$$(2\mathbf{D}_j - I)\mathbf{X}\mathbf{u}_{j,k} \geq 0, (2\mathbf{D}_j - I)\mathbf{X}\mathbf{u}'_{j,k} \geq 0, j \in [p], k \in [K].$$

*where* $\mathbf{y}_k$ *is the $k$-th column of* $\mathbf{Y}$. *The dual problem of* (7) *is given by*

$$\max \text{ tr}(\boldsymbol{\Lambda}^T \mathbf{Y}),$$

$$\text{s.t. } \mathbf{diag}(\mathbf{y}_k)\boldsymbol{\lambda}_k \succeq 0, \max_{\|\mathbf{w}\|_2 \leq 1} |\boldsymbol{\lambda}_k^T (\mathbf{X}^T \mathbf{w})_+| \leq 1, k \in [K]. \tag{8}$$

*where* $\boldsymbol{\lambda}_k$ *is the $k$-th column of* $\boldsymbol{\Lambda}$.

We present the detailed derivation of the convex formulation (7) and its dual problem (8) in the appendix. Given $\mathbf{u} \in \mathbb{R}^d$, we define $\mathbf{D}(\mathbf{u}) = \mathbf{diag}(\mathbb{I}(\mathbf{Xu} > 0))$. For two vectors $\mathbf{u}, \mathbf{v} \in \mathbb{R}^d$, we define the cosine angle between $\mathbf{u}$ and $\mathbf{v}$ by $\cos \angle(\mathbf{u}, \mathbf{v}) = \frac{\mathbf{u}^T \mathbf{v}}{\|\mathbf{u}\|_2 \|\mathbf{v}\|_2}$.

## 2.1 OUR CONTRIBUTIONS

The following theorem illustrate that for neurons satisfying $\mathbf{sign}(\mathbf{y}_k^T (\mathbf{Xw}_{1,i}^{(k)})_+) = \mathbf{sign}(w_{2,i}^{(k)})$ at initialization, $\mathbf{w}_{1,i}^{(k)}$ align to the direction of $\pm \mathbf{X}^T \mathbf{D}(\mathbf{w}_{1,i}^{(k)}) \mathbf{y}_k$ at a certain time $T$, depending on $\mathbf{sign}(w_{2,i_{k,+}}^{(k)})$ at initialization. In Section 2.3, we show that these are dual extreme points of (7).

**Theorem 1** *Consider the $K$-class classification training problem* (4) *for any dataset. Suppose that the neural network is scaled at initialization such that* $\|\mathbf{w}_{1,i}^{(k)}\|_2 = |w_{2,i}^{(k)}|$ *for* $i \in [m]$ *and* $k \in [K]$. *Assume that at initialization, for* $k \in [K]$, *there exists neurons* $(\mathbf{w}_{1,i_k}^{(k)}, \mathbf{w}_{2,i_k}^{(k)})$ *such that*

$$\mathbf{sign}(\mathbf{y}_k^T (\mathbf{Xw}_{1,i_k}^{(k)})_+) = \mathbf{sign}(w_{2,i_k}^{(k)}) = s, \tag{9}$$

*where* $s \in \{1, -1\}$. *Consider the subgradient flow applied to the non-convex problem* (4). *Let* $\delta \in (0, 1)$. *Suppose that the initialization is sufficiently close to the origin. Then, for* $k \in [K]$, *there exist* $T = T(\delta, k)$ *such that*

$$\cos \angle \left( \mathbf{w}_{1,i_k}^{(k)}(T), s \mathbf{X}^T \mathbf{D}(\mathbf{w}_{1,i_k}^{(k)}(T)) \mathbf{y}_k \right) \geq 1 - \delta.$$

Next, we impose conditions on the dataset to prove a stronger global convergence results on the flow. We say that the dataset $(\mathbf{X}, \bar{\mathbf{y}})$ is orthogonal separable among multiple classes if for all $n, n' \in [N]$,

$$\mathbf{x}_n^T \mathbf{x}_{n'} > 0, \text{ if } \bar{y}_n = \bar{y}_{n'},$$
$$\mathbf{x}_n^T \mathbf{x}_{n'} \leq 0, \text{ if } \bar{y}_n \neq \bar{y}_{n'}.$$

For orthogonal separable dataset among multiple classes, the subgradient flow for the non-convex problem (4) can find the global optimum of (5) up to a scaling constant.

**Theorem 2** *Suppose that* $(\mathbf{X}, \bar{\mathbf{y}}) \in \mathbb{R}^{N \times d} \times [K]^N$ *is orthogonal separable among multiple classes. Consider the non-convex subgradient flow applied to the non-convex problem* (4). *Suppose that the initialization is sufficiently close to the origin and scaled as in Theorem 1. Then, the non-convex subgradient flow converges to the global optimum of the convex program* (7) *and hence the non-convex objective* (5) *up to scaling.*

Therefore, the above result characterizes the *implicit regularization* of unregularized gradient flow as *convex regularization*, i.e., group $\ell_1$ norm, in the convex formulation (7). It is remarkable that group sparsity is enforced by small initialization magnitude with no explicit form of regularization.

## 2.2 CONVEX GEOMETRY OF NEURAL GRADIENT FLOW

Suppose that $\boldsymbol{\lambda} \in \mathbb{R}^N$. Here we provide an interesting geometric interpretation behind the formula

$$\cos \angle(\mathbf{u}, \mathbf{X}^T \mathbf{D}(\mathbf{u}) \boldsymbol{\lambda}) > 1 - \delta.$$

which describes a *dual extreme point* to which hidden neurons approach to as predicted by Theorem 1. We now explain the geometric intuition behind this result. Consider an ellipsoid $\{\mathbf{Xu} : \|\mathbf{u}\|_2 \leq 1\}$. A positive extreme point of this ellipsoid along the direction $\boldsymbol{\lambda}$ is defined by $\arg\max_{\mathbf{u} : \|\mathbf{u}\|_2 \leq 1} \boldsymbol{\lambda}^T \mathbf{Xu}$, which is given by the formula $\frac{\mathbf{X}^T \boldsymbol{\lambda}}{\|\mathbf{X}^T \boldsymbol{\lambda}\|_2}$. Next, we consider the rectified ellipsoid set $\mathcal{Q} := \{(\mathbf{Xu})_+ : \|\mathbf{u}\|_2 \leq 1\}$ introduced in (Ergen & Pilanci, 2021a) and shown in Figure 1. The constraint $\max_{\mathbf{u} : \|\mathbf{u}\|_2 \leq 1} |\boldsymbol{\lambda}^T (\mathbf{Xu})_+| \leq 1$ on $\boldsymbol{\lambda}$ is equivalent to $\boldsymbol{\lambda} \in \mathcal{Q}^*$. Here $\mathcal{Q}^*$ is the absolute polar set of $\mathcal{Q}$, which appears as a constraint in the convex program (8) and is defined as the following convex set

$$\mathcal{Q}^* = \{\boldsymbol{\lambda} : \max_{\mathbf{z} \in \mathcal{Q}} |\boldsymbol{\lambda}^T \mathbf{z}| \leq 1\}. \tag{10}$$

An extreme point of this non-convex body along the direction $\boldsymbol{\lambda}$ is given by the solution of the problem

$$\max_{\mathbf{u}:\|\mathbf{u}\|_2 \le 1} \boldsymbol{\lambda}^T(\mathbf{Xu})_+ = \max_{\mathbf{D}_j \in \mathcal{P}} \max_{\mathbf{u}:\|\mathbf{u}\|_2 \le 1, (2\mathbf{D}_j - I)\mathbf{Xu} \ge 0} \boldsymbol{\lambda}^T \mathbf{D}_j \mathbf{Xu}. \tag{11}$$

Here, $(\boldsymbol{\lambda}, \mathbf{u})$ are primal-dual pairs as they appear in the convex dual program (8). First, note that a stationary point of gradient flow on the objective in (11) is given by the identity $c\mathbf{u} \in \partial_{\mathbf{u}}^\circ \boldsymbol{\lambda}^T(\mathbf{Xu})_+$ where $c$ is a constant. In particular, by picking the zero as the subgradient of $(\mathbf{x}_n^T \mathbf{u})_+$ when $\mathbf{x}_n^T \mathbf{u} = 0$,

$$\mathbf{u} = \frac{\mathbf{X}^T \mathbf{D}(\mathbf{u})\boldsymbol{\lambda}}{\|\mathbf{X}^T \mathbf{D}(\mathbf{u})\boldsymbol{\lambda}\|_2} = \frac{\sum_{n=1}^N \lambda_n \mathbf{x}_n \mathbb{I}(\mathbf{u}^T \mathbf{x}_n > 0)}{\|\sum_{n=1}^N \lambda_n \mathbf{x}_n \mathbb{I}(\mathbf{u}^T \mathbf{x}_n > 0)\|_2}. \tag{12}$$

Note that the formula $\cos \angle(\mathbf{u}, \mathbf{X}^T \mathbf{D}(\mathbf{u})\boldsymbol{\lambda}) > 1 - \delta$ appearing in Theorem 1 shows that gradient flow reaches the extreme points of projected ellipsoids $\{\mathbf{D}_j \mathbf{Xu} : \|\mathbf{u}\|_2 \le 1\}$ in the direction of $\boldsymbol{\lambda} = \mathbf{y}_k$, where $\mathbf{D}_j \in \mathcal{P}$ corresponds to a valid hyperplane arrangement. This interesting phenomenon is depicted in Figures 3 and 4. The one-dimensional spikes in Figures 1 and 3 are projected ellipsoids. Detailed setup for Figure 1 to 4 and additional experiments can be found in Appendix F.

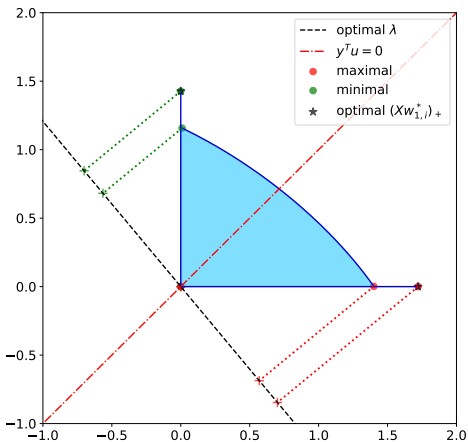

Figure 1: Rectified Ellipsoid $\mathcal{Q} := \{(\mathbf{Xu})_+ : \|\mathbf{u}\|_2 \le 1\}$ and its extreme points (spikes).

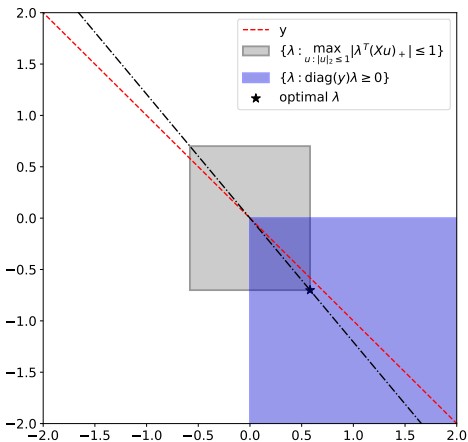

Figure 2: Convex absolute polar set $\mathcal{Q}^*$ of the Rectified Ellipsoid (purple) and other dual constraints (grey).

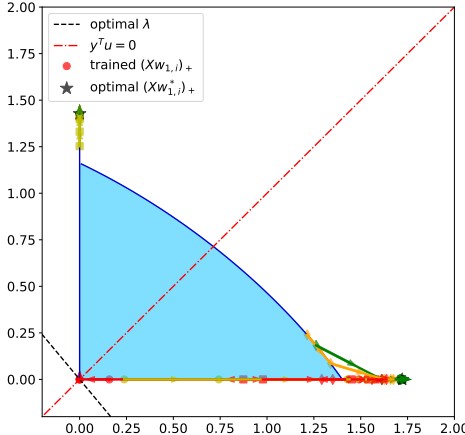

Figure 3: Trajectories of $(\mathbf{X}\hat{\mathbf{w}}_{1,i})_+$ along the training dynamics of gradient descent.

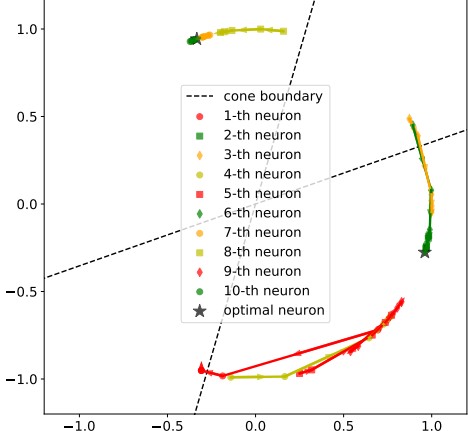

Figure 4: Trajectories of $\hat{\mathbf{w}}_{1,i} = \frac{\mathbf{w}_{1,i}}{\|\mathbf{w}_{1,i}\|_2}$ along the training dynamics of gradient descent.

Figure 5: Two-layer ReLU network gradient descent dynamics on an orthogonal separable dataset. $\hat{\mathbf{w}}_{1,i} = \frac{\mathbf{w}_{1,i}}{\|\mathbf{w}_{1,i}\|_2}$ is the normalized vector of the $i$-th hidden neuron in the first layer.

## 3 CONVEX MAX-MARGIN PROBLEM

In this section, we consider the equivalent convex model of the max-margin problem and its optimality conditions. We primarily focus on the binary classification problem for simplicity, which are later extended to the multi-class case. We can reformulate the nonconvex max-margin problem (2) as

$$\min \frac{1}{2}(\|\mathbf{W}_1\|_F^2 + \|\mathbf{w}_2\|_2^2), \text{ s.t. } \mathbf{Y}(\mathbf{X}\mathbf{W}_1)_+\mathbf{w}_2 \geq \mathbf{1}, \tag{13}$$

where $\mathbf{Y} = \mathbf{diag}(\mathbf{y})$. This is a nonconvex optimization problem due to the ReLU activation and the two-layer structure of neural network. Analogous to the convex formulation introduced in (Pilanci & Ergen, 2020) for regularized training problem of neural network, we can provide a convex optimization formulation of (13) and derive the dual problem.

**Proposition 2** *The problem* (13) *is equivalent to*

$$P_{\text{cvx}}^* = \min \sum_{j=1}^{p}(\|\mathbf{u}_j\|_2 + \|\mathbf{u}_j'\|_2),$$

$$s.t. \ \mathbf{Y} \sum_{j=1}^{p} \mathbf{D}_j \mathbf{X}(\mathbf{u}_j' - \mathbf{u}_j) \geq \mathbf{1}, \tag{14}$$

$$(2\mathbf{D}_j - I)\mathbf{X}\mathbf{u}_j \geq 0, (2\mathbf{D}_j - I)\mathbf{X}\mathbf{u}_j' \geq 0, \forall j \in [p].$$

*The dual problem of* (14) *is given by*

$$D^* = \max_{\boldsymbol{\lambda}} \mathbf{y}^T \boldsymbol{\lambda} \ s.t. \ \mathbf{Y}\boldsymbol{\lambda} \succeq 0, \ \max_{\mathbf{u}:\|\mathbf{u}\|_2 \leq 1} |\boldsymbol{\lambda}^T(\mathbf{X}^T\mathbf{u})_+| \leq 1. \tag{15}$$

The following proposition gives a characterization of the KKT point of the non-convex max-margin problem (2). The definition of $B$-subdifferential can be found in Appendix A.

**Proposition 3** *Let* $(\mathbf{W}_1, \mathbf{w}_2, \boldsymbol{\lambda})$ *be a KKT point of the non-convex max-margin problem* (2) *(in terms of B-subdifferential). Suppose that* $w_{2,i} \neq 0$ *for certain* $i \in [m]$. *Then, there exists a diagonal matrix* $\hat{\mathbf{D}}_i \in \mathbb{R}^{N \times N}$ *satisfying*

$$(\hat{\mathbf{D}}_i)_n = 1, \text{ for } \mathbf{x}_n^T\mathbf{w}_{1,i} > 0,$$

$$(\hat{\mathbf{D}}_i)_n \in \{0, 1\}, \text{ for } \mathbf{x}_n^T\mathbf{w}_{1,i} = 0,$$

$$(\hat{\mathbf{D}}_i)_n = 0, \text{ for } \mathbf{x}_n^T\mathbf{w}_{1,i} < 0.$$

*such that*

$$\frac{\mathbf{w}_{1,i}}{w_{2,i}} = \mathbf{X}^T\hat{\mathbf{D}}_i\boldsymbol{\lambda}, \|\mathbf{X}^T\hat{\mathbf{D}}_i\boldsymbol{\lambda}\|_2 = 1.$$

Based on the characterization of the KKT point of the non-convex max-margin problem (2), we provide an equivalent condition to ensure that it is also the KKT point of the convex max-margin problem (14).

**Theorem 3** *The KKT point of the non-convex max-margin problem* (13) *(in terms of B-subdifferential) corresponds to a KKT point of the convex max-margin problem* (14) *if* $\boldsymbol{\lambda}$ *is dual feasible, i.e.,*

$$\max_{\mathbf{u}:\|\mathbf{u}\|_2 \leq 1} |\boldsymbol{\lambda}^T(\mathbf{X}\mathbf{u})_+| \leq 1. \tag{16}$$

*This condition is equivalent to for all* $\mathbf{D}_j \in \mathcal{P}$, *the dual variable* $\boldsymbol{\lambda}$ *satisfies that*

$$\max_{\|\mathbf{u}\|_2 \leq 1, (2\mathbf{D}_j - I)\mathbf{X}\mathbf{u} \geq 0} |\boldsymbol{\lambda}^T\mathbf{D}_j\mathbf{X}\mathbf{u}| \leq 1. \tag{17}$$

### 3.1 DUAL FEASIBILITY OF THE DUAL VARIABLE

A natural question arises: is it possible to examine whether $\boldsymbol{\lambda}$ is feasible in the dual problem? We say the dataset $(\mathbf{X}, \mathbf{y})$ is orthogonal separable if for all $n, n' \in [N]$,

$$\mathbf{x}_n^T\mathbf{x}_{n'} > 0, \text{ if } y_n = y_{n'},$$

$$\mathbf{x}_n^T\mathbf{x}_{n'} \leq 0, \text{ if } y_n \neq y_{n'}.$$

For orthogonal separable data, as long as the induced diagonal matrices in Proposition 3 cover the positive part and the negative part of the labels, the KKT point of the non-convex max-margin problem (2) is the KKT point of the convex max-margin problem (14).

**Proposition 4** *Suppose that $(\mathbf{X}, \mathbf{y})$ is orthogonal separable. Suppose that the KKT point of the non-convex problem include two neurons $(\mathbf{w}_{1,i_+}, w_{2,i_+})$ and $(\mathbf{w}_{1,i_-}, w_{2,i_-})$ such that the corresponding diagonal matrices $\hat{\mathbf{D}}_{i_+}$ and $\hat{\mathbf{D}}_{i_-}$ defined in Proposition 3 satisfy that*

$$\hat{\mathbf{D}}_{i_+} \geq \mathbf{diag}(\mathbb{I}(y = 1)), \quad \hat{\mathbf{D}}_{i_-} \geq \mathbf{diag}(\mathbb{I}(y = -1)).$$

*Then, the dual variable $\boldsymbol{\lambda}$ is dual feasible, i.e., satisfying* (16).

The spike-free matrices discussed in (Ergen & Pilanci, 2021a) also makes examining the dual feasibility of $\boldsymbol{\lambda}$ easier. The definition of spike-free matrices can be found in Appendix A

**Proposition 5** *Suppose that $\mathbf{X}$ is spike-free. Suppose that the KKT point of the non-convex problem include two neurons $(\mathbf{w}_{1,i_+}, w_{2,i_+})$ and $(\mathbf{w}_{1,i_-}, w_{2,i_-})$ such that the corresponding diagonal matrices $\hat{\mathbf{D}}_{i_+}$ and $\hat{\mathbf{D}}_{i_-}$ defined in Proposition 3 satisfy that*

$$\hat{\mathbf{D}}_{i_+} \geq \mathbf{diag}(\mathbb{I}(y = 1)), \quad \hat{\mathbf{D}}_{i_-} \geq \mathbf{diag}(\mathbb{I}(y = -1)).$$

*Then, the dual variable $\boldsymbol{\lambda}$ is dual feasible, i.e., satisfying* (16).

**Remark 1** For the spike-free data, the constraint on the dual problem is equivalent to

$$\max_{\mathbf{Xu} \geq 0, \|\mathbf{u}\|_2 \leq 1} |\boldsymbol{\lambda}^T \mathbf{Xu}| \leq 1, \quad \text{or equivalently}$$

$$\max_{\mathbf{Xu} \geq 0, \|\mathbf{u}\|_2 \leq 1} \boldsymbol{\lambda}^T \mathbf{Y}_+ \mathbf{Xu} \leq 1, \quad \min_{\mathbf{Xu} \geq 0} \boldsymbol{\lambda}^T \mathbf{Y}_- \mathbf{Xu} \geq -1.$$

## 4 Sub-gradient flow dynamics of logistic loss

In this section, we consider the following sub-gradient flow of the logistic loss (3)

$$\frac{\partial}{\partial t}\mathbf{w}_{1,i}(t) = w_{2,i}(t)\left(\sum_{n:(\mathbf{w}_{1,i}(t))^T\mathbf{x}_n(t) > 0} \tilde{\lambda}_n(t)\mathbf{x}_n(t)\right),$$

$$\frac{\partial}{\partial t}w_{2,i}(t) = \sum_{n=1}^{N} \tilde{\lambda}_n(t)((\mathbf{w}_{1,i}(t))^T\mathbf{x}_n(t))_+. \tag{18}$$

where the $n$-th entry of $\widetilde{\boldsymbol{\lambda}}(t) \in \mathbb{R}^N$ is defined

$$\tilde{\lambda}_n = -y_n \ell'(q_n), \quad q_n = y_n(\mathbf{x}_n^T \mathbf{W}_1)_+ \mathbf{w}_2. \tag{19}$$

For simplicity, we omit the term $(t)$. For instance, we write $\mathbf{w}_{1,i} = \mathbf{w}_{1,i}(t)$. To be specific, when $\mathbf{w}_{1,i}^T \mathbf{x}_n = 0$, we select 0 as the subgradient of $w_{2,i}(\mathbf{w}_{1,i}^T \mathbf{x}_n)_+$ with respect to $\mathbf{w}_{1,i}$. Denote $\boldsymbol{\sigma}_i = \mathbf{sign}(\mathbf{Xu}_i)$. For $\boldsymbol{\sigma} \in \{1, -1, 0\}^N$, we define

$$\mathbf{g}(\boldsymbol{\sigma}, \widetilde{\boldsymbol{\lambda}}) = \sum_{n:\sigma_n > 0} \tilde{\lambda}_n \mathbf{x}_n. \tag{20}$$

For simplicity, we also write

$$\mathbf{g}(\mathbf{u}, \widetilde{\boldsymbol{\lambda}}) := \mathbf{g}(\mathbf{sign}(\mathbf{Xu}), \boldsymbol{\lambda}) = \sum_{n:\mathbf{w}_{1,i}^T\mathbf{x}_n > 0} \tilde{\lambda}_n \mathbf{x}_n. \tag{21}$$

Then, we can rewrite sub-gradient flow of the logistic loss (3) as follows:

$$\frac{\partial}{\partial t}\mathbf{w}_{i,1} = w_{2,i}\mathbf{g}(\mathbf{u}, \widetilde{\boldsymbol{\lambda}}), \quad \frac{\partial}{\partial t}w_{i,2} = \mathbf{w}_{i,1}^T\mathbf{g}(\mathbf{u}, \widetilde{\boldsymbol{\lambda}}). \tag{22}$$

Assume that the neural network is scaled at initialization, i.e., $\|\mathbf{w}_{1,i}(0)\|_2^2 = w_{2,i}^2(0)$ for $i \in [m]$. Then, the neural network is scaled for $t \geq 0$.

**Lemma 1** *Suppose that $\|\mathbf{w}_{1,i}(0)\|_2 = |\mathbf{w}_{2,i}(0)| > 0$ for $i \in [m]$. Then, for any $t > 0$, we have $\|\mathbf{w}_{1,i}(t)\|_2 = |w_{2,i}(t)| > 0$.*

According to Lemma 1, for all $t \geq 0$, $\mathbf{sign}(w_{2,i}(t)) = \mathbf{sign}(w_{2,i}(0))$. Therefore, we can simply write $s_i = s_i(t) = \mathbf{sign}(w_{2,i}(t))$. As the neural network is scaled for $t \geq 0$, it is interesting to study the dynamics of $\mathbf{w}_{1,i}$ in the polar coordinate. We write $\mathbf{w}_{1,i}(t) = e^{r_i(t)}\mathbf{u}_i(t)$, where $\|\mathbf{u}_i(t)\|_2 = 1$. The gradient flow in terms of polar coordinate writes

$$\frac{\partial}{\partial t}r_i = s_i \mathbf{u}_i^T \mathbf{g}(\mathbf{u}_i, \widetilde{\boldsymbol{\lambda}}), \quad \frac{\partial}{\partial t}\mathbf{u}_i = s_i \left( \mathbf{g}(\mathbf{u}_i, \widetilde{\boldsymbol{\lambda}}) - \left( \mathbf{u}_i^T \mathbf{g}(\mathbf{u}_i, \widetilde{\boldsymbol{\lambda}}) \right) \mathbf{u}_i \right). \tag{23}$$

Let $x_{\max} = \max_{i \in [n]} \|\mathbf{x}_i\|_2$. Define $g_{\min}$ to be

$$g_{\min} = \min_{\boldsymbol{\sigma} \in \mathcal{Q}} \|\mathbf{g}(\boldsymbol{\sigma}, \mathbf{y}/4)\|_2, \text{ s.t. } \mathbf{g}(\boldsymbol{\sigma}, \mathbf{y}/4) \neq 0, \quad \text{where we denote} \tag{24}$$

$$\mathcal{Q} = \{\boldsymbol{\sigma} \in \{1, 0, -1\}^N | \boldsymbol{\sigma} = \mathbf{sign}(\mathbf{X}\mathbf{w}), \mathbf{w} \in \mathbb{R}^d\}. \tag{25}$$

As the set $\mathcal{Q} \subseteq \{1, -1, 0\}^N$ is finite, we note that $g_{\min} > 0$. We note that when $\max_{n \in [N]} |q_n| \approx 0$, we have $\widetilde{\boldsymbol{\lambda}} \approx \frac{\mathbf{y}}{4}$. The following lemma shows that with initializations sufficiently close to 0, $\|\mathbf{g}(\mathbf{u}(t), \widetilde{\boldsymbol{\lambda}}(t)) - \mathbf{g}(\mathbf{u}(t), \mathbf{y}/4)\|_2$ and $\left\|\frac{d}{dt}\mathbf{g}(\mathbf{u}(t), \widetilde{\boldsymbol{\lambda}}(t))\right\|_2$ can be very small.

**Lemma 2** *Suppose that $T > 0$ and $\delta > 0$. Suppose that $(\mathbf{u}(t), r(t))$ follows the gradient flow (23) with $s = 1$ and the initialization $\mathbf{u}(0) = \mathbf{u}_0$ and $r(0) = r_0$. Suppose that $r_0$ is sufficiently small. Then, the following two statements hold.*

- *For all $t \leq T$, we have $\|\mathbf{g}(\mathbf{u}(t), \widetilde{\boldsymbol{\lambda}}(t)) - \mathbf{g}(\mathbf{u}(t), \mathbf{y}/4)\|_2 \leq \frac{g_{\min}\delta}{8}$.*

- *For $t \leq T$ such that $\mathbf{sign}(\mathbf{X}\mathbf{u}(s))$ is constant in a small neighbor of $t$, we have $\left\|\frac{d}{dt}\mathbf{g}(\mathbf{u}(t), \widetilde{\boldsymbol{\lambda}}(t))\right\|_2 \leq \frac{g_{\min}^2\delta}{16}$.*

Based on the above lemma on the property of $\mathbf{g}(\mathbf{u}(t), \widetilde{\boldsymbol{\lambda}}(t))$, we introduce the following lemma to upper-bound the time such that $\cos\angle(\mathbf{u}(t), \mathbf{g}(\mathbf{u}(t), \widetilde{\boldsymbol{\lambda}}(t)))$ approaches $1 - \delta$ or $\mathbf{sign}(\mathbf{X}\mathbf{u}(t))$ changes.

**Lemma 3** *Let $\delta \in (0, 1)$. Suppose that $\mathbf{u}_0$ satisfies that $\|\mathbf{u}_0\|_2 = 1$ and $\widetilde{\boldsymbol{\lambda}}(0)^T(\mathbf{X}\mathbf{u}_0)_+ > 0$. Suppose that $(\mathbf{u}(t), r(t))$ follows the gradient flow (23) with $s = 1$ and the initialization $\mathbf{u}(0) = \mathbf{u}_0$ and $r(0) = r_0$. Let $\mathbf{v}(t) = \frac{\mathbf{g}(\mathbf{u}(t), \widetilde{\boldsymbol{\lambda}}(t))}{\|\mathbf{g}(\mathbf{u}(t), \widetilde{\boldsymbol{\lambda}}(t))\|_2}$. We write $\mathbf{v}_0 = \mathbf{v}(0)$, $\boldsymbol{\sigma}_0 = \boldsymbol{\sigma}(0)$ and $g_0 = \|\mathbf{g}(\boldsymbol{\sigma}_0, \mathbf{y}/4)\|_2$. Denote*

$$T^* = \frac{1}{2g_0\sqrt{1 - \delta/8}} \left( \log \frac{\sqrt{1 - \delta/8} + 1 - \delta}{\sqrt{1 - \delta/8} - 1 + \delta} - \log \frac{\sqrt{1 - \delta/8} + \mathbf{v}_0^T \mathbf{u}_0}{\sqrt{1 - \delta/8} - \mathbf{v}_0^T \mathbf{u}_0} \right). \tag{26}$$

*For $c \in (0, 1 - \delta]$, define*

$$T^{\text{shift}}(c) = \frac{1}{2g_0\sqrt{1 - \delta/8}} \left( \log \frac{\sqrt{1 - \delta/8} + c}{\sqrt{1 - \delta/8} - c} - \log \frac{\sqrt{1 - \delta/8} + \mathbf{v}_0^T \mathbf{u}_0}{\sqrt{1 - \delta/8} - \mathbf{v}_0^T \mathbf{u}_0} \right) \tag{27}$$

*Suppose that $r_0$ is sufficiently small such that the statements in Lemma 2 holds for $T = T^*$. Then, at least one of the following event happens*

- *There exists a time $T$ such that we have $\mathbf{sign}(\mathbf{X}\mathbf{u}(t)) = \mathbf{sign}(\mathbf{X}\mathbf{u}_0)$ for $t \in [0, T)$ and $\mathbf{sign}(\mathbf{X}\mathbf{u}(t)) \neq \mathbf{sign}(\mathbf{X}\mathbf{u}_0)$. Let $\mathbf{u}_1 = \mathbf{u}(T)$ and $\mathbf{v}_1 = \lim_{t \to T - 0} \mathbf{v}(t)$. If $\mathbf{u}_1^T \mathbf{v}_1 \leq 1 - \delta$, then the time $T$ satisfies that $T \leq T^{\text{shift}}(\mathbf{v}_1^T \mathbf{u}_1)$. Otherwise, there exists a time $T'$ satisfying $T' \leq T^*$, such that we have $\mathbf{sign}(\mathbf{X}\mathbf{u}(t)) = \mathbf{sign}(\mathbf{X}\mathbf{u}_0)$ for $t \in [0, T']$ and $\mathbf{u}(T')^T \mathbf{v}(T') \geq 1 - \delta$.*

- *There exists a time $T \leq T^*$, such that we have $\mathbf{sign}(\mathbf{X}\mathbf{u}(t)) = \mathbf{sign}(\mathbf{X}\mathbf{u}_0)$ for $t \in [0, T]$ and $\mathbf{u}(T)^T \mathbf{v}(T) \geq 1 - \delta$.*

**Corollary 1** *Suppose that there exists a time $T$ such that we have $\mathbf{sign}(\mathbf{Xu}(t)) = \mathbf{sign}(\mathbf{Xu}_0)$ for $t \in [0, T)$ and $\mathbf{sign}(\mathbf{Xu}(t)) \neq \mathbf{sign}(\mathbf{Xu}_0)$. If $T > T^{\text{shift}}(\mathbf{v}_1^T \mathbf{u}_1) = \frac{1}{g_0 \sqrt{1-\delta/8}} \left( \log \frac{\sqrt{1-\delta/8} + \mathbf{v}_1^T \mathbf{u}_1}{\sqrt{1-\delta/8} - \mathbf{v}_1^T \mathbf{u}_1} - \log \frac{\sqrt{1-\delta/8} + \mathbf{v}_0^T \mathbf{u}_0}{\sqrt{1-\delta/8} - \mathbf{v}_0^T \mathbf{u}_0} \right)$, then, we have $u_1^T v_1 > 1 - \delta$.*

**Proposition 6** *Consider the sub-gradient flow (23) with $s = 1$ and the initialization $\mathbf{u}(0) = \mathbf{u}_0$ and $r(0) = r_0$. Here at initilization the neuron $\mathbf{u}_0$ satisfies that $\|\mathbf{u}_0\|_2 = 1$ and $\mathbf{y}^T(\mathbf{Xu}_0)_+ > 0$. Let $\mathbf{v}(t) = \frac{\mathbf{g}(\mathbf{u}(t), \tilde{\boldsymbol{\lambda}}(t))}{\|\mathbf{g}(\mathbf{u}(t), \tilde{\boldsymbol{\lambda}}(t))\|_2}$. For any $\delta > 0$, for sufficiently small $r_0$, there exists a time $T = \mathcal{O}(\log(\delta^{-1}))$ such that $\mathbf{u}(T)^T \mathbf{v}(T) \geq 1 - \delta$ and $\cos \angle(\mathbf{u}(T), \mathbf{g}(\mathbf{u}(T), \mathbf{y})) \geq 1 - \delta$.*

**Remark 2** The statement of proposition is similar to Lemma 4 in (Maennel et al., 2018). However, their proof contains a problem because they did not consider the change of $\mathbf{sign}(\mathbf{Xw})$ along the gradient flow. Our proof in Appendix D.4 corrects this error.

We next study the properties of orthogonal separable datasets. Denote $\mathcal{B} = \{\mathbf{w} \in \mathbb{R}^d : \|\mathbf{w}\|_2 \leq 1\}$. The following lemma give a sufficient condition on $\mathbf{w}$ to satisfy the condition in Proposition 4.

**Lemma 4** *Assume that $(\mathbf{X}, \mathbf{y})$ is orthogonal separable. Suppose that $\mathbf{w} \in \mathcal{B}$ is a local maximizer of $\mathbf{y}^T(\mathbf{Xw})_+$ in $\mathcal{B}$ and $(\mathbf{Xw})_+ \neq 0$. Then, $\langle \mathbf{w}, \mathbf{x}_n \rangle > 0$ for $n \in [N]$ such that $y_n = 1$. Suppose that $\mathbf{w} \in \mathcal{B}$ is a local minimizer of $\mathbf{y}^T(\mathbf{Xw})_+$ in $\mathcal{B}$ and $(\mathbf{Xw})_+ \neq 0$. Then, $\langle \mathbf{w}, \mathbf{x}_n \rangle > 0$ for $n \in [N]$ such that $y_n = -1$.*

We show an equivalent condition of $\mathbf{u} \in \mathcal{B}$ being the local maximizer/minimizer of $\mathbf{y}^T(\mathbf{Xu})_+$ in $\mathcal{B}$.

**Proposition 7** *Assume that $(\mathbf{X}, \mathbf{y})$ is orthogonal separable. Then, $\mathbf{u} \in \mathcal{B}$ is a local maximizer of $\mathbf{y}^T(\mathbf{Xu})_+$ in $\mathcal{B}$ is equivalent to $\cos \angle(\mathbf{u}, \mathbf{g}(\mathbf{u}, \mathbf{y})) = 1$. Similarly, $\mathbf{u} \in \mathcal{B}$ is a local minimizer of $\mathbf{y}^T(\mathbf{Xu})_+$ in $\mathcal{B}$ is equivalent to $\cos \angle(\mathbf{u}, \mathbf{g}(\mathbf{u}, \mathbf{y})) = -1$.*

Based on Proposition 4 and 7, we present the main theorem.

**Theorem 4** *Suppose that the dataset is orthogonal separable and $\boldsymbol{\theta}(t)$ follows the gradient flow. Suppose that the neural network is scaled at initialization, i.e., $\|\mathbf{w}_{1,i}(0)\|_2 = |w_{2,i}(0)|$ for all $i \in [m]$. For almost all initializations which are sufficiently close to zero, the limiting point of $\frac{\boldsymbol{\theta}(t)}{\|\boldsymbol{\theta}(t)\|_2}$ is $\frac{\boldsymbol{\theta}^*}{\|\boldsymbol{\theta}^*\|_2}$, where $\boldsymbol{\theta}^*$ is a global minimizer of the max-margin problem (2).*

We present a sketch of the proof. According to Proposition 6, for initialization sufficiently close to zero, there exist two neurons and time $T_+, T_- > 0$ such that $\cos \angle(\mathbf{w}_{1,i_+}(T_+), \mathbf{g}(\mathbf{w}_{1,i_+}(T_+), \mathbf{y})) \geq 1 - \delta$ and $\cos \angle(\mathbf{w}_{1,i_-}(T_-), \mathbf{g}(\mathbf{w}_{1,i_-}(T_-), \mathbf{y})) \leq -(1 - \delta)$. This implies that $\mathbf{w}_{1,i_+}(T_+)$ and $\mathbf{w}_{1,i_-}(T_+)$ are sufficiently close to certain stationary points of gradient flow maximizing/minimizing $\mathbf{y}^T(\mathbf{Xu}_+)$ over $\mathcal{B}$, i.e., $\{\mathbf{u} \in \mathcal{B} | \cos(\mathbf{u}, \mathbf{g}(\mathbf{u}, \mathbf{y})) = \pm 1\}$. As the dataset is orthogonal separable, from Proposition 7 and Lemma 4, the induced masking matrices $\hat{D}_{i_+}(T_+)$ and $\hat{D}_{i_-}(T_-)$ by $\mathbf{w}_{1,i_+}(T_+)/\mathbf{w}_{1,i_-}(T_-)$ in Proposition 3 satisfy that $\hat{D}_{i_+}(T_+) \geq \mathbf{diag}(\mathbb{I}(\mathbf{y} = 1))$ and $\hat{D}_{i_-}(T_-) \geq \mathbf{diag}(\mathbb{I}(\mathbf{y} = -1))$. According to Lemma 3 in (Phuong & Lampert, 2021), for $t \geq \max\{T_+, T_-\}$, we also have $\hat{D}_{i_+}(t) \geq \mathbf{diag}(\mathbb{I}(\mathbf{y} = 1))$ and $\hat{D}_{i_-}(t) \geq \mathbf{diag}(\mathbb{I}(\mathbf{y} = -1))$. According to Theorem 3 and Proposition 4, the KKT point of the non-convex problem (2) that gradient flow converges to corresponds to the KKT point of the convex problem (14).

## 5 CONCLUSION

We provide a convex formulation of the non-convex max-margin problem for two-layer ReLU neural networks and uncover a primal-dual extreme point relation between non-convex subgradient flow. Under the assumptions on the training data, we show that flows converge to KKT points of the convex max-margin problem, hence a global optimum of the non-convex objective.

## 6 ACKNOWLEDGEMENTS

This work was partially supported by the National Science Foundation under grants ECCS-2037304, DMS-2134248, and US Army Research Office.

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

## A  DEFINITIONS AND NOTIONS

We introduce several useful definitions and notions which will be utilized in the proof.

### A.1  DEFINITIONS

**Definition 1** Let $\mathcal{O} \subset \mathbb{R}^n$ be an open set and let $F : \mathcal{O} \to \mathbb{R}$ be locally Lipschitz continuous at $x \in \mathcal{O}$. Let $D_F$ be the differentiable points of $F$ in $\mathcal{O}$. The $B$-subdifferential of $F$ at $x$ is defined by

$$\partial^B F(x) := \left\{ \lim_{k \to \infty} F'(x^k) | x^k \in D_F, x_k \to x \right\}. \tag{28}$$

The set $\partial^\circ F(x) = \mathrm{co}(\partial_B F(x))$ is called Clarke's subdifferential, where co denotes the convex hull.

**Definition 2** A matrix $\mathbf{A}$ is spike-free if and only if the following conditions hold: for all $\|\mathbf{u}\|_2 \leq 1$, there exists $\|\mathbf{z}\|_2 \leq 1$ such that

$$(\mathbf{A}\mathbf{u})_+ = \mathbf{A}\mathbf{z}. \tag{29}$$

This is equivalent to say that

$$\max_{\mathbf{u}:\|\mathbf{u}\|_2 \leq 1, (I-\mathbf{X}\mathbf{X}^T)(\mathbf{X}\mathbf{u})_+ = 0} \|\mathbf{X}^\dagger(\mathbf{X}\mathbf{u})_+\|_2 \leq 1. \tag{30}$$

### A.2  NOTIONS

We use the following letters for indexing.

- The index $n$ is for the $n$-th data sample $\mathbf{x}_n$.
- We use the index $i$ to represent the $i$-th neuron-pair $(\mathbf{w}_{1,i}, w_{2,i})$.
- The index $j$ is for the $j$-th masking matrix $\mathbf{D}_i \in \mathcal{P}$.

## B  PROOFS IN SECTION 3

### B.1  PROOF FOR PROPOSITION 2

Consider the following loss function $\tilde{l} : \mathbb{R}^N \times \mathbb{R}^N \to \mathbb{R} \cup \{+\infty\}$

$$\tilde{\ell}(\mathbf{z}, \mathbf{y}) = \begin{cases} 0, & y_n z_n \geq 1, \forall n \in [N], \\ +\infty, & \text{otherwise.} \end{cases} \tag{31}$$

For a given $\mathbf{y} \in \{1, -1\}^N$, $\tilde{\ell}(\mathbf{z}, \mathbf{y})$ is a convex loss function of $\mathbf{z}$. The non-convex max-margin is equivalent to

$$\min \tilde{l}((\mathbf{X}\mathbf{W}_1)_+ \mathbf{w}_2, \mathbf{y}) + \frac{1}{2} \left( \|\mathbf{W}_1\|_F^2 + \|\mathbf{w}_2\|_2^2 \right). \tag{32}$$

According to Appendix A.13 in (Pilanci & Ergen, 2020), the problem (32) is equivalent to

$$\min \tilde{l} \left( \sum_{i=1}^p \mathbf{D}_i \mathbf{X}(\mathbf{u}_i' - \mathbf{u}_i), \mathbf{y} \right) + \frac{1}{2} \left( \|\mathbf{W}_1\|_F^2 + \|\mathbf{w}_2\|_2^2 \right),$$
$$\text{s.t. } (2\mathbf{D}_i - I)\mathbf{X}\mathbf{u}_i \geq 0, (2\mathbf{D}_i - I)\mathbf{X}\mathbf{u}_i' \geq 0, \forall i \in [p]. \tag{33}$$

This is equivalent to (14). For fixed $\mathbf{y} \in \{1, -1\}^N$, the Fenchel conjugate function of $\tilde{\ell}(\mathbf{z}, \mathbf{y})$ with respect to $\mathbf{z}$ can be computed by

$$\begin{aligned} l^*(\hat{\boldsymbol{\lambda}}, \mathbf{y}) &= \max_{\mathbf{z} \in \mathbb{R}^N} \mathbf{z}^T \mathbf{x} - \tilde{\ell}(\mathbf{z}, \mathbf{y}) \\ &= \max_{\mathbf{z} \in \mathbb{R}^N} \mathbf{z}^T \hat{\boldsymbol{\lambda}}, \text{ s.t. } \mathbf{diag}(\mathbf{y})\mathbf{z} \geq \mathbf{1}, \\ &= \begin{cases} \mathbf{y}^T \hat{\boldsymbol{\lambda}}, & \mathbf{diag}(\mathbf{y})\hat{\boldsymbol{\lambda}} \leq 0 \\ +\infty, & \text{otherwise.} \end{cases} \end{aligned} \tag{34}$$

According to Theorem 6 in (Pilanci & Ergen, 2020), the dual problem of (14) writes

$$\max -\tilde{l}^*(\boldsymbol{\lambda}, \mathbf{y}), \text{ s.t. } \max_{\mathbf{u}:\|\mathbf{u}\|_2 \leq 1} |\boldsymbol{\lambda}^T(\mathbf{Xu})_+| \leq 1, \tag{35}$$

which is equivalent to

$$\max -\mathbf{y}^T\boldsymbol{\lambda}, \text{ s.t. } \mathbf{diag}(\mathbf{y})\boldsymbol{\lambda} \leq 0, \max_{\mathbf{u}:\|\mathbf{u}\|_2 \leq 1} |\boldsymbol{\lambda}^T(\mathbf{Xu})_+| \leq 1. \tag{36}$$

By taking $\boldsymbol{\lambda} = -\hat{\boldsymbol{\lambda}}$, we derive (15). This completes the proof.

## B.2 PROOF FOR PROPOSITION 3

For the non-convex max-margin problem (13), consider the Lagrange function

$$L(\mathbf{W}_1, \mathbf{w}_2, \boldsymbol{\lambda}) = \frac{1}{2}(\|\mathbf{W}_1\|_F^2 + \|\mathbf{w}_2\|_2^2) - (\mathbf{Y}\boldsymbol{\lambda})^T(\mathbf{Y}(\mathbf{XW}_1)_+\mathbf{w}_2 - \mathbf{1})$$

where $\mathbf{Y}\boldsymbol{\lambda} \succeq 0$. The KKT point of the non-convex max-margin problem (13) (in terms of B-subdifferential) satisfies

$$\begin{aligned}
0 &\in \partial_{\mathbf{W}_1}^B L(\mathbf{W}_1, \mathbf{w}_2, \boldsymbol{\lambda}), \\
\mathbf{w}_2 &- (\mathbf{XW}_1)_+^T\boldsymbol{\lambda} = 0, \\
\lambda_n(\mathbf{y}_n(\mathbf{x}_n^T\mathbf{W}_1)_+\mathbf{w}_2 - 1) &= 0.
\end{aligned} \tag{37}$$

The KKT condition on the $i$-th column of $\mathbf{W}_1$ is equivalent to

$$\mathbf{w}_{1,i} = \sum_{n=1}^N w_{2,i}\lambda_n\mathbf{x}_n g_{n,i}, \tag{38}$$

where $g_{n,i} \in \partial^B(z)_+|_{z=\mathbf{x}_n^T\mathbf{w}_{1,i}}$. In other words, we have

$$g_{n,i} \begin{cases} = \mathbb{I}(\mathbf{x}_n^T\mathbf{w}_{1,i} \geq 0), & \text{if } \mathbf{x}_n^T\mathbf{w}_{1,i} \neq 0, \\ \in \{0,1\}, & \text{if } \mathbf{x}_n^T\mathbf{w}_{1,i} = 0. \end{cases} \tag{39}$$

Let $\hat{\mathbf{D}}_i = \mathbf{diag}([g_{1,i}, \ldots, g_{N,i}])$. Then, we can write that

$$\begin{aligned}
\mathbf{w}_{1,i} &= \sum_{n=1}^N \lambda_n g_{n,i}\mathbf{x}_n w_{2,i} \\
&= w_{2,i}\mathbf{X}^T\hat{\mathbf{D}}_i\boldsymbol{\lambda}.
\end{aligned} \tag{40}$$

From the definition of $g_{n,i}$, we have

$$g_{n,i}\mathbf{x}_n^T\mathbf{w}_{1,i} = 0. \tag{41}$$

Therefore, we can compute that

$$\begin{aligned}
w_{2,i} &= (\mathbf{Xw}_{1,i})_+^T\boldsymbol{\lambda} \\
&= \sum_{n=1}^N \mathbb{I}(\mathbf{x}_n^T\mathbf{w}_{1,i} \geq 0)\mathbf{x}_n^T\mathbf{w}_{1,i}\lambda_n \\
&= \sum_{n=1}^N g_{n,i}\mathbf{x}_n^T\mathbf{w}_{1,i}\lambda_n \\
&= \mathbf{w}_{1,i}^T\mathbf{X}^T\hat{\mathbf{D}}_i\boldsymbol{\lambda}.
\end{aligned} \tag{42}$$

In summary, we have

$$\mathbf{w}_{1,i} = w_{2,i}\mathbf{X}^T\hat{\mathbf{D}}_i\boldsymbol{\lambda}, \quad w_{2,i} = \mathbf{w}_{1,i}^T\mathbf{X}^T\hat{\mathbf{D}}_i\boldsymbol{\lambda}. \tag{43}$$

Suppose that $w_{2,i} \neq 0$. This implies that

$$\frac{\mathbf{w}_{1,i}}{w_{2,i}} = \mathbf{X}^T\hat{\mathbf{D}}_i\boldsymbol{\lambda}, \quad \|\mathbf{X}^T\hat{\mathbf{D}}_i\boldsymbol{\lambda}\|_2 = 1. \tag{44}$$

This completes the proof.

### B.3 PROOF FOR THEOREM 3

PROOF We can write the Lagrange function for the convex max-margin problem (14) as

$$L(\{\mathbf{u}_j\}_{j=1}^p, \{\mathbf{u}_j'\}_{j=1}^p, \boldsymbol{\lambda}, \{\mathbf{z}_j\}_{j=1}^p, \{\mathbf{z}_i'\}_{j=1}^p)$$

$$= \sum_{j=1}^p (\|\mathbf{u}_j\|_2 + \|\mathbf{u}_j'\|_2) + \boldsymbol{\lambda}^T \mathbf{diag}(\mathbf{y}) \left( \mathbf{1} - \mathbf{diag}(\mathbf{y}) \sum_{j=1}^p \mathbf{D}_j \mathbf{X}(\mathbf{u}_j - \mathbf{u}_j') \right)$$

$$- \sum_{j=1}^p (\mathbf{z}_j^T (2\mathbf{D}_j - I)\mathbf{X}\mathbf{u}_j + (\mathbf{z}_j')^T (2\mathbf{D}_j - I)\mathbf{X}\mathbf{u}_j') \tag{45}$$

$$= \boldsymbol{\lambda}^T \mathbf{y} + \sum_{j=1}^p (\|\mathbf{u}_j\|_2 + \|\mathbf{u}_j'\|_2) + \sum_{j=1}^p (\mathbf{u}_j')^T (\mathbf{X}^T \mathbf{D}_j \boldsymbol{\lambda} - \mathbf{X}^T (2\mathbf{D}_j - I)\mathbf{z}_j')$$

$$+ \sum_{j=1}^p (\mathbf{u}_j)^T (-\mathbf{X}^T \mathbf{D}_j \boldsymbol{\lambda} - \mathbf{X}^T (2\mathbf{D}_j - I)\mathbf{z}_j).$$

where $\mathbf{z}_j, \mathbf{z}_j' \in \mathbb{R}^N$ satisfies that $\mathbf{z}_j \geq 0, \mathbf{z}_j' \geq 0$ for $j \in [p]$ and $\boldsymbol{\lambda} \in \mathbb{R}^N$ satisfies that $\mathbf{diag}(\mathbf{y})\boldsymbol{\lambda} \geq 0$. The KKT point shall satisfy the following KKT conditions:

$$-\mathbf{X}^T \mathbf{D}_j \boldsymbol{\lambda} + \mathbf{X}^T (2\mathbf{D}_j - I)\mathbf{z}_j' \in \partial_{\mathbf{u}_j'} \|\mathbf{u}_j'\|_2,$$

$$\mathbf{X}^T \mathbf{D}_j \boldsymbol{\lambda} + \mathbf{X}^T (2\mathbf{D}_j - I)\mathbf{z}_j \in \partial_{\mathbf{u}_j} \|\mathbf{u}_j\|_2,$$

$$\lambda_n \left( \sum_{j=1}^p (\mathbf{D}_j)_n \mathbf{x}_n^T (\mathbf{u}_j - \mathbf{u}_j') - y_n \right) = 0, \tag{46}$$

$$z_{j,n}(2(\mathbf{D}_j)_{n,n} - 1)\mathbf{x}_n^T \mathbf{u}_j = 0,$$

$$z_{j,n}'(2(\mathbf{D}_j)_{n,n} - 1)\mathbf{x}_n^T \mathbf{u}_j' = 0.$$

Let $(\mathbf{W}_1, \mathbf{w}_2, \boldsymbol{\lambda})$ be the KKT point of the non-convex problem (2) and $\boldsymbol{\lambda}$ satisfies (17). Let $\hat{\mathbf{D}}_i$ be the diagonal matrix defined in Proposition 3 with respect to $\mathbf{w}_{1,i}$ and denote $\bar{\mathcal{P}} = \{\hat{\mathbf{D}}_i | i \in [m]\}$. Without the loss of generality, we may assume that $\{\bar{D}_i\}_{i=1}^m$ are different. (Otherwise, we can merge two neurons $\mathbf{w}_{1,i_1}$ and $\mathbf{w}_{1,i_2}$ with $\bar{\mathbf{D}}_{i_1} = \bar{\mathbf{D}}_{i_2}$ together.)

Suppose that $\mathbf{D}_j \in \hat{\mathcal{P}}$, i.e., $\mathbf{D}_j = \hat{\mathbf{D}}_i$ for certain $i \in [m]$. By letting $\mathbf{u}_j' = \mathbf{w}_{1,i} w_{2,i}$, $\mathbf{z}_j' = 0$, $\mathbf{u}_j = -\mathbf{w}_{1,i} w_{2,i}$ and $\mathbf{z}_j = 0$, the following identities hold.

$$\mathbf{X}^T \mathbf{D}_j \boldsymbol{\lambda} + \mathbf{X}^T (2\mathbf{D}_i - I)\mathbf{z}_j' = \mathbf{X}^T \hat{\mathbf{D}}_i \boldsymbol{\lambda} = \frac{\mathbf{w}_{1,i}}{w_{2,i}} = \frac{\mathbf{u}_i'}{\|\mathbf{u}_i'\|}. \tag{47}$$

$$-\mathbf{X}^T \mathbf{D}_j \boldsymbol{\lambda} + \mathbf{X}^T (2\mathbf{D}_i - I)\mathbf{z}_j = -\mathbf{X}^T \hat{\mathbf{D}}_i \boldsymbol{\lambda} = \frac{\mathbf{w}_{1,i}}{w_{2,i}} = \frac{\mathbf{u}_i}{\|\mathbf{u}_i\|}. \tag{48}$$

Therefore, for index $j$ satisfying $\mathbf{D}_i \in \hat{P}$, the first two KKT conditions in (46) hold.

For $\mathbf{D}_j \notin \hat{\mathcal{P}}$, we can let $\mathbf{u}_j = \mathbf{u}_j' = 0$. As $\boldsymbol{\lambda}$ satisfies (17), we have

$$\max_{\|\mathbf{u}\|_2 \leq 1, (2\mathbf{D}_j - I)\mathbf{X}\mathbf{u} \geq 0} |\boldsymbol{\lambda}^T \mathbf{D}_j \mathbf{X}\mathbf{u}| \leq 1. \tag{49}$$

According to Lemma 4 in (Pilanci & Ergen, 2020), this implies that there exist $\mathbf{z}_j, \mathbf{z}_j' \geq 0$ such that

$$\| -\mathbf{X}^T \mathbf{D}_j \boldsymbol{\lambda} + \mathbf{Z}^T (2\mathbf{D}_j - I)\mathbf{z}_j' \| \leq 1, \|\mathbf{X}^T \mathbf{D}_j \boldsymbol{\lambda} + \mathbf{Z}^T (2\mathbf{D}_j - I)\mathbf{z}_j\| \leq 1. \tag{50}$$

Therefore, the first two KKT conditions in (46) hold.

From our choice of $\mathbf{u}_j, \mathbf{z}_j, \mathbf{u}_j', \mathbf{z}_j$, the last two KKT conditions in (46) hold. We also note that

$$\sum_{j=1}^p \mathbf{D}_j \mathbf{X}(\mathbf{u}_j' - \mathbf{u}_j) = \sum_{i=1}^m (\mathbf{X}\mathbf{w}_{1,i})_+ w_{2,i}. \tag{51}$$

As $(\mathbf{W}_1, \mathbf{w}_2, \boldsymbol{\lambda})$ is the KKT point of the non-convex problem, the third KKT condition in (46) holds. This completes the proof.

## C  PROOFS IN SECTION 3.1

In this section, we present several proofs for propositions in Section 3.1.

### C.1  PROOF FOR PROPOSITION 4

We start with two lemmas.

**Lemma 5** *Suppose that* $\mathbf{u}_0 = \mathbf{X}^T \hat{\mathbf{D}}_0 \boldsymbol{\lambda}$ *and* $\|\mathbf{u}_0\|_2 \leq 1$. *For any masking matrix* $\mathbf{D}_j \in \mathcal{P}$ *such that* $(\mathbf{D}_j - \hat{\mathbf{D}}_0)\mathbb{I}(\boldsymbol{\lambda} > 0) = 0$, *we have*

$$\max_{(2\mathbf{D}_j - I)\mathbf{X}\mathbf{u} \geq 0, \|\mathbf{u}\|_2 \leq 1} \boldsymbol{\lambda}^T \mathbf{D}_j \mathbf{X} \mathbf{u} \leq 1. \tag{52}$$

PROOF  According to Lemma 4 in (Pilanci & Ergen, 2020), the constraint (52) is equivalent to that there exist $\mathbf{z}_j \in \mathbb{R}^N$ such that $\mathbf{z}_j \geq 0$ and

$$\|\mathbf{X}^T \mathbf{D}_j \boldsymbol{\lambda} + \mathbf{X}^T (2\mathbf{D}_j - I)\mathbf{z}_j\| \leq 1. \tag{53}$$

Consider the index $n \in [N]$ such that $(\mathbf{D}_j - \hat{\mathbf{D}}_0)_{nn} \neq 0$. As $(\mathbf{D}_j - \hat{\mathbf{D}}_0)\mathbb{I}(\boldsymbol{\lambda} > 0) = 0$, we have $\lambda_n \leq 0$. We let $(\mathbf{z}_j)_n = -\lambda_n$. If $(\hat{\mathbf{D}}_0)_{nn} = 0$, then we have $(\mathbf{D}_j)_{nn} = 1$ and

$$(\mathbf{D}_j - \hat{\mathbf{D}}_0)_{nn} \lambda_n \mathbf{x}_n = \lambda_n \mathbf{x}_n = -\mathbf{x}_n^T (2(\mathbf{D}_j)_{nn} - 1)(\mathbf{z}_j)_n. \tag{54}$$

If $(\hat{\mathbf{D}}_0)_{nn} = 1$, then we have $(\mathbf{D}_j)_{nn} = 0$ and

$$(\mathbf{D}_j - \hat{\mathbf{D}}_0)_{nn} \lambda_n \mathbf{x}_n = -\lambda_n \mathbf{x}_n = -\mathbf{x}_n^T (2(\mathbf{D}_j)_{nn} - 1)(\mathbf{z}_j)_n. \tag{55}$$

For other index $n \in [N]$, we simply let $(\mathbf{z}_j)_n = 0$. Then, we have

$$(\mathbf{D}_j - \hat{\mathbf{D}}_0)_{nn} \lambda_n \mathbf{x}_n = 0 = -\mathbf{x}_n^T (2(\mathbf{D}_j)_{nn} - 1)(\mathbf{z}_j)_n. \tag{56}$$

Based on our choice of $\mathbf{z}_j$, we have $\mathbf{z}_j \geq 0$ and for $n \in [N]$

$$(\mathbf{D}_j - \hat{\mathbf{D}}_0)_{nn} \lambda_n \mathbf{x}_n = -\mathbf{x}_n^T (2(\mathbf{D}_j)_{nn} - 1)(\mathbf{z}_j)_n. \tag{57}$$

This implies that

$$\mathbf{X}^T (\mathbf{D}_j - \hat{\mathbf{D}}_0)\boldsymbol{\lambda} = -\mathbf{X}^T (2\mathbf{D}_j - I)\mathbf{z}_j. \tag{58}$$

Hence, we have

$$\mathbf{X}^T \mathbf{D}_j \boldsymbol{\lambda} + \mathbf{X}^T (2\mathbf{D}_j - I)\mathbf{z}_j = \mathbf{X}^T \hat{\mathbf{D}}_0 \boldsymbol{\lambda} = \mathbf{u}_0. \tag{59}$$

Therefore, $\|\mathbf{X}^T \mathbf{D}_j \boldsymbol{\lambda} + \mathbf{X}^T (2\mathbf{D}_j - I)\mathbf{z}_j\|_2 \leq 1$.

**Lemma 6** *Suppose that the data is orthogonal separable and* $\mathbf{Y}\boldsymbol{\lambda} \geq 0$. *Suppose that* $\mathbf{u}_0 = \mathbf{X}^T \hat{\mathbf{D}}_0 \boldsymbol{\lambda}$ *and* $\|\mathbf{u}_0\|_2 \leq 1$. *For any masking matrix* $\mathbf{D}_j$ *such that* $\hat{\mathbf{D}}_0 - \mathbf{D}_j \geq 0$, *we have* $\|\mathbf{X}^T \mathbf{D}_j \boldsymbol{\lambda}\|_2 \leq \|\mathbf{u}_0\|_2 \leq 1$. *Therefore,* (52) *holds.*

PROOF  We note that $\mathbf{u}_0 = \mathbf{X}^T (\hat{\mathbf{D}}_0 - \mathbf{D}_j)\boldsymbol{\lambda} + \mathbf{X}\mathbf{D}_j \boldsymbol{\lambda}$. Denote $\mathbf{a} = \mathbf{X}^T (\hat{\mathbf{D}}_0 - \mathbf{D}_j)\boldsymbol{\lambda}$ and $\mathbf{b} = \mathbf{X}^T \mathbf{D}_j \boldsymbol{\lambda}$. We note that

$$\mathbf{a}^T \mathbf{b} = \left( \sum_{n:(\hat{\mathbf{D}}_0)_{nn}=1,(\mathbf{D}_j)_{n,n}=0} \lambda_n \mathbf{x}_n \right)^T \left( \sum_{n':(\hat{\mathbf{D}}_0)_{n'n'}=0,(\mathbf{D}_j)_{n'n'}=0} \lambda_{n'} \mathbf{x}_{n'} \right). \tag{60}$$

As $\mathbf{diag}(\mathbf{y})\boldsymbol{\lambda} \geq 0$, $\lambda_n$ has the same signature with $y_n$. Therefore, from the orthogonal separability of the data, we have

$$\lambda_n \lambda_{n'} \mathbf{x}_n^T \mathbf{x}_{n'}^T \geq 0. \tag{61}$$

This immediately implies that $\mathbf{a}^T \mathbf{b} \geq 0$. Therefore,

$$1 \geq \|\mathbf{u}_0\|_2^2 = \|\mathbf{a} + \mathbf{b}\|_2^2 = \|\mathbf{a}\|_2^2 + \|\mathbf{b}\|_2^2 + 2\mathbf{a}^T \mathbf{b} \geq \|\mathbf{a}\|_2. \tag{62}$$

This completes the proof.

Based on Lemma 5 and Lemma 6, we present the proof for Proposition 3. Let $\mathbf{u}_+ = \frac{\mathbf{w}_{1,i_+}}{w_{2,i_+}}$. From the proof of Proposition 3, we note that $\|\mathbf{u}_+\|_2 = 1$. For any masking matrix $\mathbf{D}_j \in \mathcal{P}$, let $\tilde{\mathbf{D}} = \hat{\mathbf{D}}_{i_+} \mathbf{D}_j$. As $\hat{\mathbf{D}}_{i_+} \geq \tilde{\mathbf{D}}$, according to Lemma 6, we have

$$\|\mathbf{X}^T \tilde{\mathbf{D}} \boldsymbol{\lambda}\|_2 \leq \|\mathbf{X}^T \hat{\mathbf{D}}_{i_+} \boldsymbol{\lambda}\|_2 = \|\mathbf{u}_+\|_2 \leq 1. \tag{63}$$

As $\mathbf{Y}\boldsymbol{\lambda} \geq 0$ and $\hat{\mathbf{D}}_{i_+} \geq \mathbf{diag}(\mathbb{I}(y = 1))$, we have $(\mathbf{D}_j - \tilde{\mathbf{D}})\mathbb{I}(\boldsymbol{\lambda} > 0) = \mathbf{D}_j(I - \hat{\mathbf{D}}_{i_+})\mathbb{I}(\boldsymbol{\lambda} > 0) = 0$. From Lemma 5, we note that $\boldsymbol{\lambda}$ satisfies (52). Similarly, we can show that $-\boldsymbol{\lambda}$ also satisfies (52). This completes the proof.

## C.2 Proof for Proposition 5

PROOF Note that $\mathbf{Y}\boldsymbol{\lambda} \geq 0$. Let $\mathbf{Y}_+ = \mathbf{diag}(\mathbb{I}(y=1))$ and $\mathbf{Y}_- = \mathbf{diag}(\mathbb{I}(y=-1))$. We claim that

$$\max_{\|\mathbf{u}\| \leq 1} \boldsymbol{\lambda}^T (\mathbf{Xu})_+ = \max_{\|\mathbf{u}\| \leq 1} \boldsymbol{\lambda}^T \mathbf{Y}_+ (\mathbf{Xu})_+. \tag{64}$$

Firstly, we note that

$$\boldsymbol{\lambda}^T (\mathbf{Xu})_+ = \sum_{n=1}^{N} \lambda_n (\mathbf{x}_n^T \mathbf{u})_+ \leq \sum_{n=1}^{N} (\lambda_n)_+ (\mathbf{x}_n^T \mathbf{u})_+ = \boldsymbol{\lambda}^T \mathbf{Y}_+ (\mathbf{Xu})_+. \tag{65}$$

This implies that $\max_{\|\mathbf{u}\| \leq 1} \boldsymbol{\lambda}^T (\mathbf{Xu})_+ \leq \max_{\|\mathbf{u}\| \leq 1} \boldsymbol{\lambda}^T \mathbf{Y}_+ (\mathbf{Xu})_+$.

On the other hand, suppose that $\mathbf{u} \in \arg\max_{\|\mathbf{u}\| \leq 1} \boldsymbol{\lambda}^T \mathbf{Y}_+ (\mathbf{Xu})_+$. As $\mathbf{X}$ is spike-free, there exists $\mathbf{z}$ such that $\|\mathbf{z}\|_2 \leq 1$ and $\mathbf{Xz} = (\mathbf{Xu})_+$. Therefore, we have

$$\boldsymbol{\lambda}^T \mathbf{Y}_+ (\mathbf{Xu})_+ = \boldsymbol{\lambda}^T \mathbf{Y}_+ \mathbf{Xz} = \boldsymbol{\lambda}^T \mathbf{Xz} = \boldsymbol{\lambda}^T (\mathbf{Xz})_+. \tag{66}$$

This implies that $\max_{\|\mathbf{u}\| \leq 1} \boldsymbol{\lambda}^T (\mathbf{Xu})_+ \geq \max_{\|\mathbf{u}\| \leq 1} \boldsymbol{\lambda}^T \mathbf{Y}_+ (\mathbf{Xu})_+$.

For any $\mathbf{D}_j \in \mathcal{P}$ with $\mathbf{D}_j \geq \mathbf{Y}_+$. We note that

$$\boldsymbol{\lambda}^T (\mathbf{Xu})_+ \leq \boldsymbol{\lambda}^T \mathbf{D}_j (\mathbf{Xu})_+ \leq \boldsymbol{\lambda}^T \mathbf{Y}_+ (\mathbf{Xu})_+. \tag{67}$$

Combining with (65), this implies that $\max_{\|\mathbf{u}\| \leq 1} \boldsymbol{\lambda}^T (\mathbf{Xu})_+ = \max_{\|\mathbf{u}\| \leq 1} \boldsymbol{\lambda}^T \mathbf{D}(\mathbf{Xu})_+$.

Let us go back to the original problem. Let $\mathbf{u}_+ = \frac{\mathbf{w}_{1,i_+}}{w_{2,i_+}}$. We note that $(\mathbf{Xw}_+) = \hat{\mathbf{D}}_{i_+} \mathbf{Xw}_+ = \hat{\mathbf{D}}_{i_+} \mathbf{XX}^T \hat{\mathbf{D}}_{i_+} \boldsymbol{\lambda}$. Therefore, we have

$$\boldsymbol{\lambda}^T (\mathbf{Xw}_+) = \boldsymbol{\lambda}^T \hat{\mathbf{D}}_{i_+} \mathbf{XX}^T \hat{\mathbf{D}}_{i_+} \boldsymbol{\lambda} = \|\mathbf{X}^T \hat{\mathbf{D}}_{i_+} \boldsymbol{\lambda}\|_2^2 = \|\mathbf{u}_+\|_2^2 = 1. \tag{68}$$

Thus, for any $\|\mathbf{u}\|_2 \leq 1$, suppose that $(\mathbf{Xu})_+ = \mathbf{Xz}$, where $\|\mathbf{z}\|_2 \leq 1$. Then, we have

$$\boldsymbol{\lambda}^T \hat{\mathbf{D}}_{i_+} (\mathbf{Xu})_+ = \boldsymbol{\lambda}^T \hat{\mathbf{D}}_{i_+} \mathbf{Xz} \leq \|\mathbf{z}\|_2 \leq 1. \tag{69}$$

Therefore, $\max_{\|\mathbf{u}\| \leq 1} \boldsymbol{\lambda}^T (\mathbf{Xu})_+ = \max_{\|\mathbf{u}\| \leq 1} \boldsymbol{\lambda}^T \mathbf{D}_+ (\mathbf{Xu})_+ \leq 1$. Similarly, we have

$$\min_{\|\mathbf{u}\| \leq 1} \boldsymbol{\lambda}^T (\mathbf{Xu})_+ = \min_{\|\mathbf{u}\| \leq 1} \boldsymbol{\lambda}^T \mathbf{D}_- (\mathbf{Xu})_+ \geq -1.$$

This completes the proof.

# D Proofs in Section 4

## D.1 Proof for Lemma 1

PROOF According to the sub-gradient flow (22), we can compute that

$$\frac{\partial}{\partial t} \left( \|\mathbf{w}_{1,i}\|_2^2 - w_{2,i}^2 \right) = 2\mathbf{w}_{1,i}^T \left( w_{2,i} \mathbf{g}(\mathbf{u}, \widetilde{\boldsymbol{\lambda}}) \right) - 2w_{2,i} \mathbf{w}_{1,i}^T \mathbf{g}(\mathbf{u}, \widetilde{\boldsymbol{\lambda}}) = 0. \tag{70}$$

Let $T_0 = \sup\{T | \|\mathbf{w}_{1,i}(t)\|_2 = |w_{2,i}(t)| > 0, \forall i \in [n], t \in [0,T)\}$. For $t \in [0, T_0)$, as the neural network is scaled, it is sufficient study the dynamics of $\mathbf{w}_{1,i}$ in the polar coordinate. Let us write $\mathbf{w}_{1,i}(t) = e^{r_i(t)} \mathbf{u}_i(t)$, where $\|\mathbf{u}_i(t)\|_2 = 1$. Then, in terms of polar coordinate, the projected gradient flow follows

$$\begin{aligned}
\frac{\partial}{\partial t} r_i &= \mathbf{sign}(w_{2,i}) \mathbf{u}_i^T \mathbf{g}(\mathbf{u}_i, \widetilde{\boldsymbol{\lambda}}), \\
\frac{\partial}{\partial t} \mathbf{u}_i &= \mathbf{sign}(w_{2,i}) \left( \mathbf{g}(\mathbf{u}_i, \widetilde{\boldsymbol{\lambda}}) - \left( \mathbf{u}_i^T \mathbf{g}(\mathbf{u}_i, \widetilde{\boldsymbol{\lambda}}) \right) \mathbf{u}_i \right).
\end{aligned} \tag{71}$$

Without the loss of generality, we may assume that $w_{2,i}(0) \neq 0$ for $i \in [m]$. Denote

$$x_{\max} = \max_{i \in [n]} \|\mathbf{x}_i\|_2. \tag{72}$$

From the definition of $\widetilde{\boldsymbol{\lambda}}$, we have $\|\widetilde{\boldsymbol{\lambda}}\|_\infty \leq 1/4$. Therefore, we have

$$\left|\frac{\partial}{\partial t} r_i\right| \leq \|\mathbf{g}(\mathbf{u}_i, \widetilde{\boldsymbol{\lambda}})\|_2 \leq \left\|\sum_{j:\mathbf{x}_j^T \mathbf{u}>0} \tilde{\lambda}_j \mathbf{x}_j\right\|_2 \leq \frac{n x_{\max}}{2}. \tag{73}$$

Therefore, for finite $t > 0$, we have

$$r_i(t) \geq r_i(0) - \frac{n x_{\max} t}{4}, \tag{74}$$

which implies that $|w_{2,i}(t)| > 0$. This implies that $T_0 = \infty$.

## D.2 Proof of Lemma 2

PROOF As we have $\|\mathbf{w}_{1,i}\|_2 = |w_{2,i}|$, for $n \in [N]$, we can compute that

$$\begin{aligned}
|q_n| &= |(\mathbf{x}_n^T \mathbf{W}_1)_+ \mathbf{w}_2| \\
&\leq \sum_{i=1}^m |(\mathbf{x}_n^T \mathbf{w}_{1,i})_+ w_{2,i}| \\
&= \sum_{i=1}^m \|\mathbf{w}_{1,i}\|_2 |(\mathbf{x}_n^T \mathbf{w}_{1,i})_+| \\
&\leq \sum_{i=1}^m \|\mathbf{w}_{1,i}\|_2 |\mathbf{x}_n^T \mathbf{w}_{1,i}| \\
&\leq \|\mathbf{x}_n\|_2 \sum_{i=1}^m \|\mathbf{w}_{1,i}\|_2^2.
\end{aligned} \tag{75}$$

Note that $\tilde{\lambda}_n = -y_n \ell'(q_n)$ and $\frac{y_n}{4} = -y_n \ell'(0)$. As $\ell'$ is $\frac{1}{4}$-Lipschitz continuous, we have

$$\left|\tilde{\lambda}_n - y_n/4\right| \leq \frac{1}{4}|q_n| \leq \frac{\|\mathbf{x}_n\|_2}{4} \sum_{i=1}^m \|\mathbf{w}_{1,i}\|_2^2. \tag{76}$$

For any $\hat{\boldsymbol{\sigma}} \in \mathcal{Q}$, as $\tilde{\lambda}_n \in [0, 1/4]$ for $n \in [N]$, we have

$$\begin{aligned}
&\|\mathbf{g}(\hat{\sigma}, \widetilde{\boldsymbol{\lambda}}) - \mathbf{g}(\hat{\sigma}, \mathbf{y}/4)\|_2 \\
&\leq \left\|\sum_{n:(\hat{\boldsymbol{\sigma}})_n>0} (\tilde{\lambda}_n - y_n/4)\mathbf{x}_n\right\|_2 \\
&\leq \sum_{k:(\hat{\sigma})_k>0} |\lambda_k - \tilde{\lambda}_k| \|\mathbf{x}_k\|_2 \\
&\leq \sum_{k=1}^n \frac{\|\mathbf{x}_k\|_2^2}{4} \sum_{j=1}^m \|\mathbf{w}_{1,j}\|_2^2 \\
&= c_1 \sum_{i=1}^m \|\mathbf{w}_{1,i}\|_2^2,
\end{aligned} \tag{77}$$

where $c_1 = \frac{1}{4}\|\mathbf{X}\|_F^2 > 0$ is a constant. Therefore, we can bound $\|\mathbf{g}(\hat{\sigma}, \widetilde{\boldsymbol{\lambda}}(t))\|$ by

$$\|\mathbf{g}(\hat{\sigma}, \widetilde{\boldsymbol{\lambda}}(t))\|_2 \leq \|\mathbf{g}(\hat{\sigma}, \mathbf{y}/4)\|_2 + c_1 \sum_{i=1}^m \|\mathbf{w}_{1,i}(t)\|_2^2 \leq d_{\max} + c_1 \sum_{i=1}^m e^{2r_i(t)}, \tag{78}$$

where we let

$$g_{\max} = \max_{\boldsymbol{\sigma} \in \mathcal{Q}} \|\mathbf{g}(\sigma, \mathbf{y}/4)\|_2. \tag{79}$$

Let $r(t) = \max_{i \in [m]} r_i(t)$. We note that

$$\frac{\partial}{\partial t} r(t) \leq d_{\max} + c_1 n e^{2r(t)} \leq c_2(1 + e^{2r(t)}), \tag{80}$$

where $c_2 = \max\{nc_1, d_{\max}\} > 0$ is a constant. If we start with $r(0) \ll 0$, then, $r(t)$ cannot grow much faster than $c_2 t$. Let $\tilde{r}(t)$ satisfy the following ODE:

$$\frac{\partial}{\partial t} \tilde{r}(t) = c_2(1 + e^{2\tilde{r}(t)}). \tag{81}$$

The solution is given by

$$\tilde{r}_a(t) = c_2(t - a) - \frac{1}{2} \log(1 - e^{2c_2(t-a)}), \tag{82}$$

where $a > 0$ is a parameter depending on the initialization. For any initial $r(0)$, we have a unique $a$ satisfying $\tilde{r}_a(0) = r(0)$. Therefore, we have $r(t) \leq \tilde{r}_a(t)$ and

$$\|\mathbf{g}(\sigma, \widetilde{\boldsymbol{\lambda}}(t)) - \mathbf{g}(\sigma, \mathbf{y}/4)\|_2 \leq c_1 n e^{2\tilde{r}_a(t)}. \tag{83}$$

According to the bound (83), by choosing a sufficiently small $r_0$, (which leads to a sufficiently small $a$), such that

$$e^{2\tilde{r}_a(T)} \leq \min\left\{\frac{d_{\min}\delta}{16c_1}, \frac{d_{\min}^2\delta}{4n^2 x_{\max}^3}\right\}. \tag{84}$$

Therefore, for $t \leq T$, we have

$$\|\mathbf{g}(\hat{\boldsymbol{\sigma}}, \widetilde{\boldsymbol{\lambda}}(t)) - \mathbf{g}(\hat{\boldsymbol{\sigma}}, \mathbf{y}/4)\|_2 \leq c_1 n e^{2\tilde{r}_a(t)} \leq c_1 n e^{2\tilde{r}_a(T)} \leq \frac{d_{\min}\delta}{8}. \tag{85}$$

Hence, we have

$$\|\mathbf{g}(\boldsymbol{\sigma}(t), \widetilde{\boldsymbol{\lambda}}(t)) - \mathbf{g}(\boldsymbol{\sigma}(t), \mathbf{y}/4)\|_2 \leq c_1 n e^{2\tilde{r}_a(t)} \leq c_1 n e^{2\tilde{r}_a(T)} \leq \frac{d_{\min}\delta}{8}. \tag{86}$$

We can compute that

$$\frac{d}{dt}\tilde{\lambda}_i = -y_i l^{(2)}(q_i)\frac{d}{dt}q_i. \tag{87}$$

As $l^{(2)}(q) \in (0, 1/4]$, we can compute that

$$\begin{aligned}
\left|\frac{d}{dt}q_i\right| &\leq \sum_{j=1}^{m} |w_{2,j}| \|\mathbf{x}_i\|_2 \left\|\frac{d}{dt}\mathbf{w}_{1,j}\right\|_2 \\
&\quad + \sum_{j=1}^{m} \|\mathbf{w}_{1,j}\|_2 \|\mathbf{x}_i\|_2 \left|\frac{d}{dt}w_{2,j}\right| \\
&\leq \frac{n}{4}\sum_{j=1}^{m} \|\mathbf{w}_{1,j}\|_2^2 x_{\max}^2 + \frac{n}{4}\sum_{j=1}^{m} w_{2,j}^2 x_{\max}^2 \\
&\leq \frac{n x_{\max}^2}{2} e^{2r(t)}.
\end{aligned} \tag{88}$$

Therefore, we have

$$\left|\frac{d}{dt}\tilde{\lambda}_i\right| = |\ell''(q_i)| \left|\frac{d}{dt}q_i\right| \leq \frac{1}{4}\left|\frac{d}{dt}q_i\right| \leq \frac{n x_{\max}^2}{8} e^{2r(t)}. \tag{89}$$

Suppose that $\mathbf{sign}(\mathbf{X}\mathbf{u}(s)) = \boldsymbol{\sigma}(t)$ holds for $s$ in a small neighbor of $t$. Then, we have

$$\begin{aligned}
\left\|\frac{d}{dt}\mathbf{g}(\mathbf{u}(t), \widetilde{\boldsymbol{\lambda}}(t))\right\|_2 &= \left\|\frac{d}{dt}\mathbf{g}(\boldsymbol{\sigma}(t), \widetilde{\boldsymbol{\lambda}}(t))\right\|_2 \\
&\leq \sum_{i=1}^{n} \|\mathbf{x}_i\|_2 \left|\frac{d}{dt}\tilde{\lambda}_i\right| \leq \frac{n^2 x_{\max}^3}{8} e^{2r(t)} \\
&\leq \frac{n^2 x_{\max}^3}{8} e^{2\tilde{r}_a(T)} \leq \frac{d_{\min}^2\delta}{16}.
\end{aligned} \tag{90}$$

This completes the proof.

### D.3 Proof of Lemma 3

PROOF Let $T_0 = \sup\{T|\mathbf{sign}(\mathbf{X}\mathbf{u}(t)) = \mathbf{sign}(\mathbf{X}\mathbf{u}_0), \forall t \in [0, T)\}$. We analyze the dynamics of $\mathbf{u}(t)$ in the interval $[0, \min\{T_0, T^*\}]$. For $t \leq \min\{T_0, T^*\}$, as the statements in Lemma 2 hold, we can compute that

$$
\begin{aligned}
&\frac{d}{dt}\mathbf{v}(t)^T\mathbf{u}(t) \\
&= \frac{d}{dt}\left(\frac{\mathbf{g}(\boldsymbol{\sigma}_0, \widetilde{\boldsymbol{\lambda}}(t))}{\|\mathbf{g}(\boldsymbol{\sigma}_0, \widetilde{\boldsymbol{\lambda}}(t))\|_2}^T \mathbf{u}(t)\right) \\
&= \left(\frac{\mathbf{g}(\boldsymbol{\sigma}_0, \widetilde{\boldsymbol{\lambda}}(t))}{\|\mathbf{g}(\boldsymbol{\sigma}_0, \widetilde{\boldsymbol{\lambda}}(t))\|_2}\right)^T \frac{d}{dt}\mathbf{u}(t) + \mathbf{u}(t)^T \frac{1}{\|\mathbf{g}(\boldsymbol{\sigma}_0, \widetilde{\boldsymbol{\lambda}}(t))\|_2}\frac{d}{dt}\mathbf{g}(\boldsymbol{\sigma}_0, \widetilde{\boldsymbol{\lambda}}(t)) \\
&\quad - \mathbf{u}(t)^T\mathbf{g}(\boldsymbol{\sigma}_0, \widetilde{\boldsymbol{\lambda}}(t))\frac{\mathbf{g}(\boldsymbol{\sigma}_0, \widetilde{\boldsymbol{\lambda}}(t))^T\frac{d}{dt}\mathbf{g}(\boldsymbol{\sigma}_0, \widetilde{\boldsymbol{\lambda}}(t))}{\|\mathbf{g}(\boldsymbol{\sigma}_0, \widetilde{\boldsymbol{\lambda}}(t))\|_2^3} \\
&\geq \mathbf{g}(\boldsymbol{\sigma}_0, \widetilde{\boldsymbol{\lambda}}(t))^T\frac{d}{dt}\mathbf{u}_t - \frac{2}{g_{\min}}\left\|\frac{d}{dt}\mathbf{g}(\boldsymbol{\sigma}_0, \widetilde{\boldsymbol{\lambda}}(t))\right\|_2 \\
&\geq \|\mathbf{g}(\boldsymbol{\sigma}_0, \widetilde{\boldsymbol{\lambda}}(t))\|_2\left(1 - (\mathbf{v}(t)^T\mathbf{u}(t))^2\right) - \frac{g_{\min}\delta}{8} \\
&\geq g_0\left(1 - \delta/8\right)\left(1 - (\mathbf{v}(t)^T\mathbf{u}(t))^2\right) - \frac{g_{\min}\delta}{8} \\
&\geq g_0\left(1 - \delta/4 - (\mathbf{v}(t)^T\mathbf{u}(t))^2\right).
\end{aligned}
\tag{91}
$$

Here we utilize that $g_0 \geq g_{\min}$, where $g_{\min}$ is defined in (24). Let $z(t)$ satisfies the ODE

$$
\frac{dz(t)}{dt} = (1 - \delta/4 - z(t)^2)g_0,
\tag{92}
$$

with initialization $z(0) = \mathbf{v}_0^T\mathbf{u}_0$. Then, we note that

$$
z(t) = \sqrt{1 - \delta/8} - \frac{2\sqrt{1 - \delta/8}}{1 + c_3\exp(2g_0t/(\sqrt{1 - \delta/8}))},
\tag{93}
$$

where $c_3 = \left(\frac{\sqrt{1-\delta/8}}{\mathbf{v}_0^T\mathbf{u}_0} - 1\right)^{-1}$. We can compute that

$$
z(T_3) = 1 - \delta.
\tag{94}
$$

According to the comparison theorem, for $t \leq \min\{T_0, T_3\}$, we have

$$
\mathbf{v}(t)^T\mathbf{u}(t) \geq z(t).
\tag{95}
$$

We first consider the case where $T_0 = \infty$. As $T_0 = \infty$, we have

$$
\mathbf{v}(T^*)^T\mathbf{u}(T^*) \geq z(T^*) = 1 - \delta.
\tag{96}
$$

Therefore, the second event holds for $T \leq T^*$.

Otherwise, we have $T_0 < \infty$. Recall that $\mathbf{u}_1 = \mathbf{u}(T_0)$ and $\mathbf{v}_1 = \lim_{t\uparrow T_0}\mathbf{v}(t)$. Let $T_1 = \sup\{T|\mathbf{v}(t)^T\mathbf{u}(t) < \mathbf{v}_1^T\mathbf{u}_1, \forall t \in [0, T)\}$ and $T_2 = \sup\{T|\mathbf{v}(t)^T\mathbf{u}(t) < 1 - \delta, \forall t \in [0, T)\}$. If $T_2 \leq T_0$, for $t \in [0, T_2]$, we have

$$
\frac{d}{dt}\mathbf{v}(t)^T\mathbf{u}(t) \geq \left(1 - \delta/4 - (1 - \delta)^2\right)g_0 > 0.
\tag{97}
$$

Therefore, $\mathbf{v}(t)^T\mathbf{u}(t)$ monotonically increases in $[0, T_2]$. As $\mathbf{v}(t)^T\mathbf{u}(t) \geq z(t)$ for $t \in [0, T_0]$, we have that $z(T_2) \leq \mathbf{v}(T_2)^T\mathbf{u}(T_2) = 1 - \delta = z(T_3)$. Hence, we have $T_2 \leq T^*$. Therefore, the second condition of the first event holds at $T = T_2$.

Then, we consider the case where $T_2 \geq T_0$. For $t \leq T_0$, we have $\mathbf{v}(t)^T\mathbf{u}(t) \leq 1 - \delta$. This implies that $\mathbf{v}_1^T\mathbf{u}_1 \leq 1 - \delta$. Apparently, we have $T_1 \leq T_0$. If $T_1 < T_0$, as $T_0 \leq T_2$, for $t \in [0, T_0]$, the inequality

(97) holds. This implies that $\lim_{t \to T_0 - 0} \mathbf{v}(t)^T \mathbf{u}(T_0) > \mathbf{v}(T_1)^T \mathbf{u}(T_1) = \lim_{t \to T_0 - 0} \mathbf{v}(t)^T \mathbf{u}(T_0)$, which leads to a contradiction. Therefore, we have $T_0 = T_1$. We note that

$$z(T^{\text{shift}}(\mathbf{u}_1^T \mathbf{v}_1)) = \mathbf{u}_1^T \mathbf{v}_1. \tag{98}$$

As $\mathbf{u}(t)^T \mathbf{g}(\mathbf{u}(t), \widetilde{\boldsymbol{\lambda}}(t)) \geq z(t)$ for $t \in [0, T_0]$, we have that $z(T_1) \leq \mathbf{u}_1^T \mathbf{v}_1 \leq z(T_4)$. Hence, we have $T_0 = T_1 \leq T^{\text{shift}}(\mathbf{u}_1^T \mathbf{v}_1)$. This completes the proof.

### D.4  PROOF OF PROPOSITION 6

We first introduce a lemma.

**Lemma 7** *Let* $\mathbf{a}, \mathbf{b} \in \mathbb{R}^d$ *and* $0 < \delta < c$. *Suppose that* $\|\mathbf{a} - \mathbf{b}\|_2 \leq \delta$ *and* $\|\mathbf{a}\|_2 \geq c$. *Then, we have*

$$\left\| \frac{\mathbf{a}}{\|\mathbf{a}\|_2} - \frac{\mathbf{b}}{\|\mathbf{b}\|_2} \right\|_2 \leq \frac{2\delta}{c}. \tag{99}$$

PROOF As $\delta < c$, we have $\|b\|_2 > \|a\|_2 - \|a - b\|_2 \geq c - \delta > 0$. We first note that

$$\left| \|\mathbf{a}\|_2^{-1} - \|\mathbf{b}\|_2^{-1} \right| = \frac{\left| \|\mathbf{a}\|_2 - \|\mathbf{b}\|_2 \right|}{\|\mathbf{a}\|_2 \|\mathbf{b}\|_2} \leq \frac{\delta}{c \|\mathbf{b}\|_2}. \tag{100}$$

Therefore, we can compute that

$$
\begin{aligned}
&\left\| \frac{\mathbf{a}}{\|\mathbf{a}\|_2} - \frac{\mathbf{b}}{\|\mathbf{b}\|_2} \right\|_2 \\
&\leq \left\| \frac{\mathbf{a}}{\|\mathbf{a}\|_2} - \frac{\mathbf{b}}{\|\mathbf{a}\|_2} \right\|_2 + \left| \frac{1}{\|\mathbf{a}\|_2} - \frac{1}{\|\mathbf{b}\|_2} \right| \|\mathbf{b}\|_2 \\
&\leq \frac{\delta}{c} + \frac{\delta}{c} = \frac{2\delta}{c}.
\end{aligned}
\tag{101}
$$

This completes the proof.

Then we present the proof of Proposition 6.

PROOF As $\mathbf{y}^T(\mathbf{X}\mathbf{u}_0)_+ > 0$, with sufficiently small initialization and sufficiently small $\delta > 0$, we also have $\widetilde{\boldsymbol{\lambda}}(0)^T(\mathbf{X}\mathbf{u}_0)_+ \geq \mathbf{y}^T(\mathbf{X}\mathbf{u}_0)_+/4 - \|\mathbf{X}\|_2 \|\widetilde{\boldsymbol{\lambda}}(0) - \mathbf{y}/4\|_2 > 0$. We prove that there exists a time $T$ such that $\mathbf{u}(T)^T \mathbf{v}(T) \geq 1 - \frac{3}{4}\delta$ by contradiction. Denote $\mathbf{v}_0 = \mathbf{v}(0)$. For all possible values of $\|\mathbf{g}(\mathbf{u}, \mathbf{y}/4)\|_2$, we can arrange them from the smallest to the largest by $g_{(1)} < g_{(2)} < \cdots < g_{(p)}$. Let $T_i = \frac{1}{2\sqrt{1 - \delta/8} g_{(i)}} \left( \log \frac{\sqrt{1 - \delta/8} + 1 - \delta/2}{\sqrt{1 - \delta/8} - 1 + \delta/2} - \log \frac{\sqrt{1 - \delta/8} + g_{(i)}^{-1} \mathbf{v}_0^T \mathbf{u}_0}{\sqrt{1 - \delta/8} - g_{(i)}^{-1} \mathbf{v}_0^T \mathbf{u}_0} \right)$ and $T = \sum_{i=1}^p T_i$. Suppose that $r_0$ is sufficiently small such that statements in Lemma 2 holds for $T$. According to Lemma 3, we can find $0 = t_0 < t_1 < \ldots$ such that for $i = 1, \ldots, \text{sign}(\mathbf{X}\mathbf{u}(t))$ is constant on $[t_{i-1}, t_i)$ and $\text{sign}(\mathbf{X}\mathbf{u}(t_{i-1})) \neq \text{sign}(\mathbf{X}\mathbf{u}(t_i))$. We write $\mathbf{u}_i = \mathbf{u}(t_i)$, $g_i = \|\mathbf{g}(\mathbf{u}(t_i), \mathbf{y}/4)\|_2$,

$$\mathbf{g}_i^- = \lim_{t \uparrow t_i} \mathbf{g}(\mathbf{u}(t), \widetilde{\boldsymbol{\lambda}}(t)), \quad \mathbf{g}_i = \mathbf{g}(\mathbf{u}(t_i), \widetilde{\boldsymbol{\lambda}}(t_i)), \tag{102}$$

$\mathbf{v}_i^- = \frac{\mathbf{g}_i^-}{\|\mathbf{g}_i^-\|_2}$ and $\mathbf{v}_i = \frac{\mathbf{g}_i}{\|\mathbf{g}_i\|_2}$. We note that $\mathbf{g}_i^- = \mathbf{g}_{i-1}^+$. According to Lemma 3, we have

$$
\begin{aligned}
t_i - t_{i-1} &\leq \frac{1}{2\sqrt{1 - \delta/8} g_{i-1}} \left( \log \frac{\sqrt{1 - \delta/8} + (\mathbf{v}_i^-)^T \mathbf{u}_i}{\sqrt{1 - \delta/8} - (\mathbf{v}_i^-)^T \mathbf{u}_i} - \log \frac{\sqrt{1 - \delta/8} + \mathbf{v}_{i-1}^T \mathbf{u}_{i-1}}{\sqrt{1 - \delta/8} - \mathbf{v}_{i-1}^T \mathbf{u}_{i-1}} \right) \\
&\leq \frac{1}{2\sqrt{1 - \delta/8} g_{\min}} \left( \log \frac{\sqrt{1 - \delta/8} + (\mathbf{v}_i^-)^T \mathbf{u}_i}{\sqrt{1 - \delta/8} - (\mathbf{v}_i^-)^T \mathbf{u}_i} - \log \frac{\sqrt{1 - \delta/8} + \mathbf{v}_{i-1}^T \mathbf{u}_{i-1}}{\sqrt{1 - \delta/8} - \mathbf{v}_{i-1}^T \mathbf{u}_{i-1}} \right).
\end{aligned}
\tag{103}
$$

Here we utilize that $g_{i-1} \geq g_{\min}$, where $g_{\min}$ is defined in (24). This implies that

$$\frac{\sqrt{1 - \delta/8} + (\mathbf{v}_i^-)^T \mathbf{u}_i}{\sqrt{1 - \delta/8} - (\mathbf{v}_i^-)^T \mathbf{u}_i} \geq e^{2\sqrt{1 - \delta/8} g_{\min}(t_i - t_{i-1})} \frac{\sqrt{1 - \delta/8} + \mathbf{v}_{i-1}^T \mathbf{u}_{i-1}}{\sqrt{1 - \delta/8} - \mathbf{v}_{i-1}^T \mathbf{u}_{i-1}}. \tag{104}$$

We can show that for $t$ satisfying $t \geq \frac{1}{2\sqrt{1-\delta/8}g_{\min}}\left(\log\frac{\sqrt{1-\delta/8}+1-\delta/2}{\sqrt{1-\delta/8}-1+\delta/2} - \log\frac{1+g_{\min}^{-1}\mathbf{v}_0^T\mathbf{u}_0}{1-g_{\min}^{-1}\mathbf{v}_0^T\mathbf{u}_0}\right)$ and $t \leq T$, we have $\|\mathbf{g}(\mathbf{u}(t),\boldsymbol{\lambda})\|_2 > g_{\min}$. According to Lemma 3, as $g_i \geq g_{\min}$, we have

$$\frac{\sqrt{1-\delta/8} + g_{\min}^{-1}(\mathbf{g}_i^-)^T\mathbf{u}_i}{\sqrt{1-\delta/8} - g_{\min}^{-1}(\mathbf{g}_i^-)^T\mathbf{u}_i} \geq e^{2\sqrt{1-\delta/8}g_{\min}(t_i-t_{i-1})}\frac{\sqrt{1-\delta/8} + g_{\min}^{-1}\mathbf{g}_{i-1}^T\mathbf{u}_{i-1}}{\sqrt{1-\delta/8} - g_{\min}^{-1}\mathbf{g}_{i-1}^T\mathbf{u}_{i-1}}. \tag{105}$$

This implies that

$$\frac{\sqrt{1-\delta/8} + g_{\min}^{-1}(\mathbf{g}_i^-)^T\mathbf{u}_i}{\sqrt{1-\delta/8} - g_{\min}^{-1}(\mathbf{g}_i^-)^T\mathbf{u}_i} \geq e^{2\sqrt{1-\delta/8}g_{\min}t_i}\frac{\sqrt{1-\delta/8} + g_{\min}^{-1}\mathbf{v}_0^T\mathbf{u}_0}{\sqrt{1-\delta/8} - g_{\min}^{-1}\mathbf{v}_0^T\mathbf{u}_0}, \tag{106}$$

or equivalently, for any $t > 0$, we have

$$\frac{\sqrt{1-\delta/8} + g_{\min}^{-1}(\mathbf{g}(\mathbf{u}(t)),\widetilde{\boldsymbol{\lambda}}(t))^T\mathbf{u}(t)}{\sqrt{1-\delta/8} - g_{\min}^{-1}\mathbf{g}(\mathbf{u}(t),\widetilde{\boldsymbol{\lambda}}(t))^T\mathbf{u}(t)} \geq e^{2\sqrt{1-\delta/8}g_{\min}t}\frac{\sqrt{1-\delta/8} + g_{\min}^{-1}\mathbf{v}_0^T\mathbf{u}_0}{\sqrt{1-\delta/8} - g_{\min}^{-1}\mathbf{v}_0^T\mathbf{u}_0}. \tag{107}$$

Here we utilize that $\mathbf{g}(\mathbf{u}(t),\boldsymbol{\lambda}(t))^T\mathbf{u}(t)$ is continuous w.r.t. $t$. Therefore, for $t \geq \frac{1}{2g_{\min}}\left(\log\frac{2-\delta}{\delta} - \log\frac{1+g_{\min}^{-1}\mathbf{v}_0^T\mathbf{u}_0}{1-g_{\min}^{-1}\mathbf{v}_0^T\mathbf{u}_0}\right)$, we have

$$\frac{1 + g_{\min}^{-1}(\mathbf{g}(\mathbf{u}(t),\widetilde{\boldsymbol{\lambda}}(t)))^T\mathbf{u}(t)}{1 - g_{\min}^{-1}\mathbf{g}(\mathbf{u}(t),\widetilde{\boldsymbol{\lambda}}(t))^T\mathbf{u}(t)} \geq \frac{\sqrt{1-\delta/8}+1-\delta/2}{\sqrt{1-\delta/8}-1+\delta/2}. \tag{108}$$

This implies that

$$g_{\min}^{-1}(\mathbf{g}(\mathbf{u}(t),\widetilde{\boldsymbol{\lambda}}(t)))^T\mathbf{u}(t) \geq 1 - \delta/2. \tag{109}$$

If $\|\mathbf{g}(\mathbf{u}(t),\boldsymbol{\lambda})\|_2 = g_{\min}$, as the statements in Lemma 2 hold, we can compute that

$$\|\mathbf{g}(\mathbf{u}(t),\boldsymbol{\lambda}) - \mathbf{g}(\mathbf{u}(t),\widetilde{\boldsymbol{\lambda}}(t))\|_2 \leq \frac{g_{\min}\delta}{4} = \frac{\delta}{4}\|\mathbf{g}(\mathbf{u}(t),\boldsymbol{\lambda})\|_2, \tag{110}$$

which implies that

$$\|\mathbf{g}(\mathbf{u}(t),\widetilde{\boldsymbol{\lambda}}(t))\|_2 \leq (1 + \delta/4)\|\mathbf{g}(\mathbf{u}(t),\boldsymbol{\lambda})\|_2. \tag{111}$$

Therefore, we have

$$\begin{aligned}
\mathbf{v}(t)^T\mathbf{u}(t) &= \frac{(\mathbf{g}(\mathbf{u}(t),\widetilde{\boldsymbol{\lambda}}(t)))^T\mathbf{u}(t)}{\|\mathbf{g}(\mathbf{u}(t),\widetilde{\boldsymbol{\lambda}}(t))\|_2} \\
&\geq \frac{1}{1+\delta/4}\frac{(\mathbf{g}(\mathbf{u}(t),\widetilde{\boldsymbol{\lambda}}(t)))^T\mathbf{u}(t)}{g_{\min}} \\
&\geq \frac{1-\delta/2}{1+\delta/4} \geq 1 - \frac{3}{4}\delta.
\end{aligned} \tag{112}$$

This leads to a contradiction.

Analogously, we can show that for $t \geq \sum_{j=1}^i T_i$, we have $\|\mathbf{g}(\mathbf{u}(t),\mathbf{y}/4)\|_2 > g_{(i)}$. Thus, by taking $t \geq \sum_{i=1}^p T_i$, we have $\|\mathbf{g}(\mathbf{u}(t),\mathbf{y}/4)\|_2 > g_{(p)} = g_{\max}$. However, from the definition of $g_{\max}$, we have $\|\mathbf{g}(\mathbf{u}(t),\mathbf{y}/4)\|_2 \leq g_{\max}$. This leads to a contradiction. Therefore, there exists a time $T = \sum_{i=1}^p T_i = O(\log\delta^{-1})$ such that $\mathbf{v}(T)^T\mathbf{u}(T) \geq 1 - \frac{3}{4}\delta$.

We note that $\|\mathbf{g}(\mathbf{u}(T),\mathbf{y}/4)\|_2 \geq g_{\min}$. As the statements in Lemma (2) hold, we have

$$\|\mathbf{g}(\mathbf{u}(T),\mathbf{y}/4) - \mathbf{g}(\mathbf{u}(T),\widetilde{\boldsymbol{\lambda}}(T))\|_2 \leq \frac{\delta g_{\min}}{8} \tag{113}$$

According to Lemma 7, we have

$$\left\|\frac{\mathbf{g}(\mathbf{u}(T),\mathbf{y}/4)}{\|\mathbf{g}(\mathbf{u}(T),\mathbf{y}/4)\|_2} - \frac{\mathbf{g}(\mathbf{u}(T),\widetilde{\boldsymbol{\lambda}}(T))}{\|\mathbf{g}(\mathbf{u}(T),\widetilde{\boldsymbol{\lambda}}(T))\|_2}\right\|_2 \leq \frac{2\|\mathbf{g}(\mathbf{u}(T),\mathbf{y}/4) - \mathbf{g}(\mathbf{u}(T),\widetilde{\boldsymbol{\lambda}}(T))\|_2}{g_{\min}} \leq \frac{\delta}{4}. \tag{114}$$

This implies that

$$
\begin{aligned}
&\mathbf{u}(T)^T \frac{\mathbf{g}(\mathbf{u}(T), \mathbf{y}/4)}{\|\mathbf{g}(\mathbf{u}(T), \mathbf{y}/4)\|_2} \\
&\geq \mathbf{u}(T)^T \mathbf{v}(T) - \left\| \frac{\mathbf{g}(\mathbf{u}(T), \boldsymbol{\lambda})}{\|\mathbf{g}(\mathbf{u}(T), \boldsymbol{\lambda})\|_2} - \frac{\mathbf{g}(\mathbf{u}(T), \widetilde{\boldsymbol{\lambda}}(T))}{\|\mathbf{g}(\mathbf{u}(T), \widetilde{\boldsymbol{\lambda}}(T))\|_2} \right\|_2 \\
&\geq 1 - \delta.
\end{aligned}
\tag{115}
$$

Hence, we have

$$
\cos \angle (\mathbf{u}(T), \mathbf{g}(\mathbf{u}(T), \mathbf{y})) = \mathbf{u}(T)^T \frac{\mathbf{g}(\mathbf{u}(T), \mathbf{y})}{\|\mathbf{g}(\mathbf{u}(T), \mathbf{y})\|_2} = \mathbf{u}(T)^T \frac{\mathbf{g}(\mathbf{u}(T), \mathbf{y}/4)}{\|\mathbf{g}(\mathbf{u}(T), \mathbf{y}/4)\|_2} \geq 1 - \delta.
$$

This completes the proof.

### D.5  PROOF OF LEMMA 4

PROOF  This is proved in Lemma 2 in (Phuong & Lampert, 2021). Here we provide an alternative proof. It is sufficient to prove for the case of local maximizer. Suppose that $\mathbf{w}$ is a local maximizer of $\mathbf{y}^T (\mathbf{Xw})_+$ in $\mathcal{B}$. We first consider the case where $\mathbf{y}^T (\mathbf{Xw})_+ > 0$.

If there exists $n \in [N]$ such that $\langle \mathbf{w}, \mathbf{x}_n \rangle \leq 0$ and $y_n = 1$. Consider $\mathbf{v} = \mathbf{x}_n / \|\mathbf{x}_n\|_2$ and let $\mathbf{w}_\epsilon = \frac{\mathbf{w} + \epsilon \mathbf{v}}{\|\mathbf{w} + \epsilon \mathbf{v}\|_2}$, where $\epsilon > 0$. For index $n' \in [N]$ such that $y_{n'} = 1$, as the dataset is orthogonal separable, we have $\mathbf{x}_{n'}^T \mathbf{x}_n > 0$ and

$$
\mathbf{x}_{n'}^T (\mathbf{w} + \epsilon \mathbf{v}) = \mathbf{x}_{n'}^T \mathbf{w} + \frac{\epsilon}{\|\mathbf{x}_n\|_2} \mathbf{x}_{n'}^T \mathbf{x}_n > \mathbf{x}_{n'}^T \mathbf{w}.
\tag{116}
$$

This implies that $(\mathbf{x}_{n'}^T \mathbf{w}_\epsilon)_+ \geq (\mathbf{x}_{n'}^T \mathbf{w})_+$. For $y_{n'} = -1$, as the data is orthogonal separable, we note that $\mathbf{x}_{n'}^T \mathbf{x}_n \leq 0$ and

$$
\mathbf{x}_{n'}^T (\mathbf{w} + \epsilon \mathbf{v}) = \mathbf{x}_{n'}^T \mathbf{w} + \frac{\epsilon}{\|\mathbf{x}_n\|_2} \mathbf{x}_{n'}^T \mathbf{x}_n \leq \mathbf{x}_{n'}^T \mathbf{w}.
\tag{117}
$$

This implies that $(\mathbf{x}_j^T \mathbf{w}_\epsilon)_+ \leq (\mathbf{x}_j^T \mathbf{w})_+$. In summary, we have

$$
\mathbf{y}^T (\mathbf{X}(\mathbf{w} + \epsilon \mathbf{v}))_+ = \sum_{n=1}^N y_n (\mathbf{x}_j^T (\mathbf{w} + \epsilon \mathbf{v}))_+ \geq \sum_{n=1}^N y_n (\mathbf{x}_j^T \mathbf{w})_+ = \mathbf{y}^T (\mathbf{Xw})_+ > 0
\tag{118}
$$

If $\langle \mathbf{w}, \mathbf{x}_n \rangle < 0$, then $\mathbf{w}^T \mathbf{v} < 0$. This implies that with sufficiently small $\epsilon$, we have $\|\mathbf{w} + \epsilon \mathbf{v}\|_2 < \|\mathbf{w}\|_2 = 1$. Therefore,

$$
\mathbf{y}^T (\mathbf{Xw}_\epsilon))_+ = \frac{1}{\|\mathbf{w} + \epsilon \mathbf{v}\|_2} \mathbf{y}^T (\mathbf{X}(\mathbf{w} + \epsilon \mathbf{v}))_+ > \mathbf{y}^T (\mathbf{X}(\mathbf{w} + \epsilon \mathbf{v}))_+ \geq \mathbf{y}^T (\mathbf{Xw})_+,
\tag{119}
$$

which leads to a contradiction. If $\langle \mathbf{w}, \mathbf{x}_n \rangle = 0$, we note that

$$
(\mathbf{x}_n^T (\mathbf{w} + \epsilon \mathbf{v}))_+ = \epsilon > (\mathbf{x}_n^T \mathbf{w})_+.
\tag{120}
$$

This implies that

$$
\mathbf{y}^T (\mathbf{X}(\mathbf{w} + \epsilon \mathbf{v}))_+ \geq \mathbf{y}^T (\mathbf{Xw})_+ + \epsilon.
\tag{121}
$$

We also note that $\|\mathbf{w} + \epsilon \mathbf{v}\|_2 = \sqrt{1 + \epsilon^2} = 1 + O(\epsilon^2)$. Therefore, with sufficiently small $\epsilon$, we have

$$
\mathbf{y}^T (\mathbf{Xw}_\epsilon)_+ \geq \frac{\mathbf{y}^T (\mathbf{Xw})_+ + \epsilon}{\sqrt{1 + \epsilon^2}} > \mathbf{y}^T (\mathbf{Xw})_+.
\tag{122}
$$

We then consider the case where $\mathbf{y}^T (\mathbf{Xw})_+ < 0$. Apparently, we can make $\mathbf{y}^T (\mathbf{Xw})_+$ larger by replacing $\mathbf{w}$ by $(1 - \epsilon)\mathbf{w}$, where $\epsilon \in (0, 1)$, which leads to a contradiction.

Finally, we consider the case where $\mathbf{y}^T (\mathbf{Xw})_+ = 0$. This implies that

$$
\sum_{n:y_n=1} (\mathbf{x}_j^T \mathbf{w})_+ = \sum_{n:y_n=-1} (\mathbf{x}_j^T \mathbf{w})_+.
\tag{123}
$$

As $(\mathbf{Xw})_+ \neq 0$, this implies that there exists at least for one index $n \in [N]$ such that $y_n = 1$ and $\mathbf{x}_n^T \mathbf{w} > 0$. Let $\mathbf{v} = \mathbf{x}_n / \|\mathbf{x}_n\|_2$. We note that $\frac{1}{\|\mathbf{w} + \epsilon \mathbf{v}\|_2} \mathbf{y}^T (\mathbf{X}(\mathbf{w} + \epsilon \mathbf{v}))_+ > 0$ for $\epsilon > 0$. This leads to a contradiction.

### D.6 PROOF OF PROPOSITION 7

It is sufficient to consider the case of the local maximizer. Denote $\mathcal{Q} = \{\boldsymbol{\sigma} \in \{-1, 0, 1\}^N | \mathbf{diag}(\boldsymbol{\sigma}) \in \mathcal{P}\}$. For $\boldsymbol{\sigma}, \boldsymbol{\sigma}' \in \mathcal{Q}$, we say $\boldsymbol{\sigma} \subseteq \boldsymbol{\sigma}'$ if for all index $n \in [N]$ with $\sigma_n \neq 0$, $\sigma_n' = \sigma_n$. We say $\sigma \in \mathcal{Q}$ is open if $\boldsymbol{\sigma}_n \neq 0$ for $n \in [N]$. Define

$$S_{\boldsymbol{\sigma}} = \{\mathbf{u} | \mathbf{sign}(\mathbf{X}\mathbf{u}) = \boldsymbol{\sigma}\}. \tag{124}$$

We start with the two lemmas.

**Lemma 8** *Let $\boldsymbol{\lambda} \in \mathbb{R}^N$. Suppose that $\mathbf{u}_0$ satisfies that $\mathbf{u}_0 = \frac{\mathbf{g}(\mathbf{u}_0, \boldsymbol{\lambda})}{\|\mathbf{g}(\mathbf{u}_0, \boldsymbol{\lambda})\|_2}$. Let $\boldsymbol{\sigma} = \mathbf{sign}(\mathbf{u}_0)$. Then, $\mathbf{v} \in \mathcal{B}_2$ is a local maximizer of $\boldsymbol{\lambda}^T(\mathbf{X}\mathbf{u})_+$ in $\mathcal{B}_2$ if for any open $\boldsymbol{\sigma}'$ satisfying $\boldsymbol{\sigma} \subseteq \boldsymbol{\sigma}'$, we have $\|\mathbf{g}(\boldsymbol{\sigma}, \mathbf{y})\|_2 = \|\mathbf{g}(\boldsymbol{\sigma}', \mathbf{y})\|_2$.*

PROOF Suppose that $\boldsymbol{\sigma}$ is open. Then, $S_{\boldsymbol{\sigma}}$ is an open set. In a small neighbor around $\mathbf{u}_0 = \frac{\mathbf{g}(\mathbf{u}_0, \boldsymbol{\lambda})}{\|\mathbf{g}(\mathbf{u}_0, \boldsymbol{\lambda})\|_2} = \frac{\mathbf{g}(\boldsymbol{\sigma}, \boldsymbol{\lambda})}{\|\mathbf{g}(\boldsymbol{\sigma}, \boldsymbol{\lambda})\|_2}$, $\boldsymbol{\lambda}^T(\mathbf{X}\mathbf{u})_+ = \mathbf{u}^T\mathbf{g}(\boldsymbol{\sigma}, \boldsymbol{\lambda})$ is a linear function of $\mathbf{u}$. The Riemannian gradient of $\mathbf{u}^T\mathbf{g}(\boldsymbol{\sigma}, \boldsymbol{\lambda})$ at $\mathbf{v}$ is zero. This implies that $\mathbf{v}$ locally maximizes $\boldsymbol{\lambda}^T(\mathbf{X}\mathbf{u})_+$.

Suppose that there exists at least one zero in $\boldsymbol{\sigma}$. Consider any $\mathbf{v} \in \mathcal{B}$ satisfying $\mathbf{u}_0^T\mathbf{v} = 0$. Let $\epsilon > 0$ be a small constant such that for any $s \in (0, \epsilon]$, $\mathbf{u}_0 + s\mathbf{v} \in S_{\boldsymbol{\sigma}'}$ where $\boldsymbol{\sigma} \subseteq \boldsymbol{\sigma}'$. Let $\mathbf{u}_s = \frac{\mathbf{u} + s\mathbf{v}}{\sqrt{1+s^2}}$.

Suppose that $\|\mathbf{g}(\boldsymbol{\sigma}'', \boldsymbol{\lambda})\|_2 \leq \|\mathbf{g}(\boldsymbol{\sigma}, \boldsymbol{\lambda})\|_2$ for all open $\boldsymbol{\sigma}''$ satisfying $\boldsymbol{\sigma} \subseteq \boldsymbol{\sigma}''$. For any $\boldsymbol{\sigma}'$ with $\boldsymbol{\sigma} \subseteq \boldsymbol{\sigma}'$, we construct $\boldsymbol{\sigma}''$ by $\sigma_i'' = -1$ for $n \in [N]$ such that $\sigma_n' = 0$ and $\sigma_n'' = \sigma_n'$ for $n \in [N]$ such that $\sigma_n' = 0$. We note that $\|\mathbf{g}(\boldsymbol{\sigma}'', \boldsymbol{\lambda})\|_2 \geq \|\mathbf{g}(\boldsymbol{\sigma}', \boldsymbol{\lambda})\|_2$. Thus, $\|\mathbf{g}(\boldsymbol{\sigma}', \boldsymbol{\lambda})\|_2 \leq \|\mathbf{g}(\boldsymbol{\sigma}'', \boldsymbol{\lambda})\|_2 \leq \|\mathbf{g}(\boldsymbol{\sigma}, \boldsymbol{\lambda})\|_2$. As $|\boldsymbol{\lambda}^T(\mathbf{X}\mathbf{u}_s)_+| = |\mathbf{g}(\boldsymbol{\sigma}', \boldsymbol{\lambda})^T\mathbf{u}_s| \leq \|\mathbf{g}(\boldsymbol{\sigma}', \boldsymbol{\lambda})\|_2$, we have $|\boldsymbol{\lambda}^T(\mathbf{X}\mathbf{u}_s)_+| \leq \|\mathbf{g}(\boldsymbol{\sigma}', \boldsymbol{\lambda})\|_2 \leq \|\mathbf{g}(\boldsymbol{\sigma}, \boldsymbol{\lambda})\|_2$. Therefore, $\mathbf{u}$ is a local maximizer of $\boldsymbol{\lambda}^T(\mathbf{X}\mathbf{u})_+$.

**Lemma 9** *Suppose that the dataset is orthogonal separable. Let $\boldsymbol{\lambda} \in \mathbb{R}^N$ satisfy that $\mathbf{diag}(\mathbf{y})\boldsymbol{\lambda} \geq 0$. Suppose that $\mathbf{u}_0$ satisfies that $\mathbf{u}_0 = \frac{\mathbf{g}(\mathbf{u}_0, \boldsymbol{\lambda})}{\|\mathbf{g}(\mathbf{u}_0, \boldsymbol{\lambda})\|_2}$. Then, for any $\boldsymbol{\sigma}'$ satisfying $\boldsymbol{\sigma} \subseteq \boldsymbol{\sigma}'$, we have $\|\mathbf{g}(\boldsymbol{\sigma}', \boldsymbol{\lambda})\|_2 = \|\mathbf{g}(\boldsymbol{\sigma}, \boldsymbol{\lambda})\|_2$.*

PROOF If there exists $n \in [N]$ such that $\sigma_n = 1$ and $y_n = -1$, as the data is orthogonal separable, we note that

$$\mathbf{x}_n^T\mathbf{g}(\boldsymbol{\sigma}, \boldsymbol{\lambda}) = \mathbf{x}_n^T\left(\sum_{n':\sigma_{n'}>0} \lambda_{n'}\mathbf{x}_{n'}\right) = y_n(y_n\mathbf{x}_n)^T\left(\sum_{n':\sigma_{n'}>0}(\lambda_{n'}y_{n'})y_{n'}\mathbf{x}_{n'}\right) \leq 0, \tag{125}$$

which contradicts with $\mathbf{sign}(\mathbf{x}_n^T\mathbf{g}(\boldsymbol{\sigma}, \boldsymbol{\lambda})) = \mathbf{sign}(\mathbf{x}_n^T\mathbf{u}_0) = \sigma_n = 1$.

Suppose that there exists $n \in [N]$ such that $\sigma_n$ and $y_n = 1$. Then, as the dataset is orthogonal separable, then, for index $n_1 \in [N]$ such that $\sigma_{n_1} = 0$, we note that $y_{n_1} \neq 1$. Otherwise,

$$\mathbf{x}_{n_1}^T\mathbf{g}(\boldsymbol{\sigma}, \boldsymbol{\lambda}) = \mathbf{x}_{n_1}^T\left(\sum_{n_2:\sigma_{n_2}>0} \lambda_{n_2}\mathbf{x}_{n_2}\right) = \mathbf{x}_{n_1}^T\left(\sum_{n_2:\sigma_{n_2}>0}(\lambda_{n_2}y_{n_2})y_{n_2}\mathbf{x}_{n_2}\right) > 0, \tag{126}$$

which contradicts with $\mathbf{sign}(\mathbf{x}_{n_1}^T\mathbf{g}(\boldsymbol{\sigma}, \boldsymbol{\lambda})) = \mathbf{sign}(\mathbf{x}_{n_1}^T\mathbf{u}_0) = \sigma_{n_1} = 0$. This also implies that the index set $\{n \in [N]|\sigma_n > 0\}$ include all data with $y_n = 1$.

If there exists $\boldsymbol{\sigma}'$ such that $\boldsymbol{\sigma} \subseteq \boldsymbol{\sigma}'$ and $\|\mathbf{g}(\boldsymbol{\sigma}', \boldsymbol{\lambda})\|_2 > \|\mathbf{g}(\boldsymbol{\sigma}, \boldsymbol{\lambda})\|_2$. Then, there exists at least one index $n \in [N]$ such that $\sigma_n \leq 0$ and $\sigma_n' = 1$. However, from the previous derivation, we note that $y_n = -1$ and

$$\mathbf{x}_n^T\mathbf{g}(\boldsymbol{\sigma}', \boldsymbol{\lambda}) = \mathbf{x}_n^T\left(\sum_{j:\boldsymbol{\sigma}_{n_1}'>0} \lambda_{n_1}\mathbf{x}_{n_1}\right) = \mathbf{x}_n^T\left(\sum_{n_1:\boldsymbol{\sigma}_{n_1}'>0}(\lambda_{n_1}y_{n_1})y_{n_1}\mathbf{x}_{n_1}\right) < 0, \tag{127}$$

which contradicts with $\sigma_n' = 1$.

By combining Lemma 8 and 9, we complete the proof.

### D.7 PROOF OF THEOREM 4

PROOF For almost all initialization, we can find two neurons such that $\mathbf{sign}(w_{2,i_+}) = \mathbf{sign}(\mathbf{y}^T(\mathbf{X}\mathbf{w}_{1,i_+})_+) = 1$ and $\mathbf{sign}(w_{2,i_-}) = \mathbf{sign}(\mathbf{y}^T(\mathbf{X}\mathbf{w}_{1,i_-})_+) = -1$ at initialization. By choosing a sufficiently small $\delta > 0$ in Proposition 6, there exist two neurons $\mathbf{w}_{1,i_+}, \mathbf{w}_{1,i_-}$ and times $T_+, T_- > 0$ such that $\cos\angle(\mathbf{w}_{1,i_+}(T_+), \mathbf{g}(\mathbf{w}_{1,i_+}(T_+), \mathbf{y})) > 1 - \delta$ and $\cos\angle(\mathbf{w}_{1,i_+}(T_+), \mathbf{g}(\mathbf{w}_{1,i_+}(T_+), \mathbf{y})) < -(1 - \delta)$. This implies that $\mathbf{w}_{1,i_+}(T_+)$ and $\mathbf{w}_{1,i_-}(T_+)$ are sufficiently close to certain stationary points of gradient flow maximizing/minimizing $\mathbf{y}^T(\mathbf{X}\mathbf{u}_+)$ over $\mathcal{B}$, i.e., $\{\mathbf{u} \in \mathcal{B}|\cos(\mathbf{u}, \mathbf{g}(\mathbf{u}, \mathbf{y})) = \pm 1\}$. As the dataset is orthogonal separable, according to Lemma 4 and Proposition 7, the corresponding diagonal matrices $\hat{\mathbf{D}}_{i_+}(T_+)$ and $\hat{\mathbf{D}}_{i_-}(T_-)$ satisfy that $\hat{\mathbf{D}}_{i_+}(T_+) \geq \mathbf{diag}(\mathbb{I}(\mathbf{y} = 1))$ and $\hat{\mathbf{D}}_{i_-}(T_-) \geq \mathbf{diag}(\mathbb{I}(\mathbf{y} = -1))$. According to Lemma 3 in (Phuong & Lampert, 2021), we have $\hat{\mathbf{D}}_{i_+}(t) \geq \mathbf{diag}(\mathbb{I}(\mathbf{y} = 1))$ and $\mathbf{D}_{i_-}(t) \geq \mathbf{diag}(\mathbb{I}(\mathbf{y} = -1))$ hold for $t \geq \max\{T_+, T_-\}$.

With $t \to \infty$, according to Proposition 4, the dual variable $\boldsymbol{\lambda}$ in the KKT point of the non-convex max-margin problem (13) is dual feasible, i.e., $\boldsymbol{\lambda}$ satisfies (16). Suppose that $\boldsymbol{\theta}^*$ is a limiting point of $\left\{\frac{\boldsymbol{\theta}(t)}{\|\boldsymbol{\theta}(t)\|_2}\right\}_{t \geq 0}$ and $\boldsymbol{\lambda}^*$ is the corresponding dual variable. From Theorem 1, we note that the pair $(\boldsymbol{\theta}^*, \boldsymbol{\lambda}^*)$ corresponds to the KKT point of the convex max-margin problem (14).

## E PROOFS OF MAIN RESULTS ON MULTI-CLASS CLASSIFICATION

### E.1 PROOF OF PROPOSITION 1

The neural network training problem (4) can be separated into $K$ subproblems. Each of these subproblems corresponds to the neural network training problem (3) for binary classification. For each subproblem, by applying Proposition 2, we complete the proof.

### E.2 PROOF OF THEOREM 1

We note that the neural network training problem (4) can be separated into $K$ subproblems. Each of these subproblems corresponds to the neural network training problem (3) for binary classification. By applying Proposition 6 with to each subproblem with $\mathbf{y} = \mathbf{y}_k$, we complete the proof.

### E.3 PROOF OF THEOREM 2

Similarly, the corresponding non-convex max-margin problem (5) and the convex max-margin problem (7) can be separated into $K$ subproblems. Each of these subproblems corresponds to the non-convex max-margin problem (2) and the convex max-margin problem (14) for binary classification. By applying Theorem 4 to each subproblem with $\mathbf{y} = \mathbf{y}_k$, we complete the proof.

## F NUMERICAL EXPERIMENT

### F.1 DETAILS ON FIGURE 5

We provide the experiment setting in Figure 1 and 5 as follows. The dataset is given by $\mathbf{X} = \begin{bmatrix} 1.65 & -0.47 \\ -0.47 & 1.35 \end{bmatrix} \in \mathbb{R}^{2 \times 2}$ and $\mathbf{y} = \begin{bmatrix} 1 \\ -1 \end{bmatrix} \in \mathbb{R}^2$. Here we have $N = 2$ and $d = 2$. We note that this dataset is orthogonal separable but not spike-free. We plot the ellipsoid set and the rectified ellipsoid set in Figure 6.

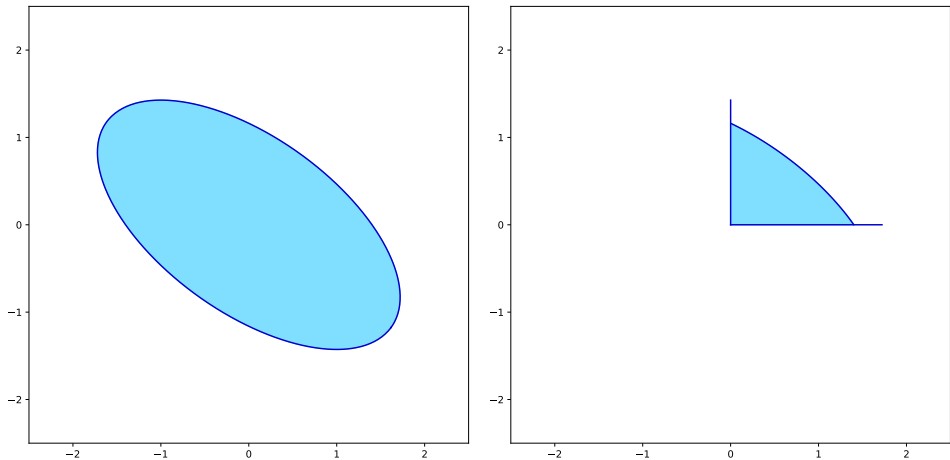

Figure 6: The ellipsoid set and the rectified ellipsoid set. Orthogonal separable dataset.

We enumerate all possible hyperplane arrangements in the set $\mathcal{P}$ and solve the convex max-margin problem (14) via CVXPY to obtain the following non-zero neurons

$$\mathbf{u}_{1,3} = \begin{bmatrix} 0.58 \\ -0.16 \end{bmatrix}, \quad \mathbf{w}'_{1,2} = \begin{bmatrix} -0.23 \\ 0.66 \end{bmatrix}. \tag{128}$$

We note that the dual problem (15) is equivalent to

$$\begin{aligned}
\max \ & \boldsymbol{\lambda}^T \mathbf{y}, \\
\text{s.t. } & \|\mathbf{X}^T \mathbf{D}_j \boldsymbol{\lambda} - \mathbf{X}^T (2\mathbf{D}_j - I)\mathbf{z}_{j,+}\|_2 \le 1, \forall j \in [p], \\
& \| - \mathbf{X}^T \mathbf{D}_j \boldsymbol{\lambda} - \mathbf{X}^T (2\mathbf{D}_j - I)\mathbf{z}_{j,-}\|_2 \le 1, \forall j \in [p], \\
& \mathbf{z}_{j,+} \ge 0, \mathbf{z}_{j,-} \ge 0, \forall j \in [p], \mathbf{diag}(\mathbf{y})\boldsymbol{\lambda} \ge 0.
\end{aligned} \tag{129}$$

The above problem is a second-order cone program (SOCP) and can be solved via standard convex optimization frameworks such as CVX and CVXPY. We solve (129) to obtain the optimal dual variable $\boldsymbol{\lambda}$. For the geometry of the dual problem, as the dataset is orthogonal separable, the set $\{\boldsymbol{\lambda} : \max_{\|\mathbf{u}\|_2 \le 1} |\boldsymbol{\lambda}^T(\mathbf{Xu})_+| \le 1\}$ reduces to $\{\boldsymbol{\lambda} : \max_{\|\mathbf{u}\|_2 \le 1} |\boldsymbol{\lambda}^T(\mathbf{Xu}_1^*)_+| \le 1, \boldsymbol{\lambda}^T(\mathbf{Xu}_2^*)_+| \le 1\}$, where $\mathbf{u}_1^*, \mathbf{u}_2^*$ correspond to two vectors at the spikes of the rectified ellipsoid set. We draw the sets $\{\boldsymbol{\lambda} : \max_{\|\mathbf{u}\|_2 \le 1} |\boldsymbol{\lambda}^T(\mathbf{Xu})_+| \le 1\}$, $\{\boldsymbol{\lambda} :$, the optimal dual variable $\boldsymbol{\lambda}$ and the direction of $\mathbf{y}$ in Figure 2.

For each $\mathbf{D}_j \in \mathcal{P}$, we solve for the vector $\mathbf{u}_j$ which maximize/minimize $\boldsymbol{\lambda}^T \mathbf{D}_j \mathbf{Xu}_j$ with the constraints $\|\mathbf{u}_j\|_2 \le 1$ and $(2\mathbf{D}_j - I)\mathbf{Xu}_j \ge 0$. We plot the rectified ellipsoid set $\{(\mathbf{Xu})_+ | \|\mathbf{u}\|_2 \le 1\}$, vectors $\mathbf{u}_j$, neurons in the optimal solution to (14) scaled to unit $\ell_2$-norm and the direction of $\boldsymbol{\lambda}$ in Figure 1. We note that each neuron $\mathbf{u}_j^*$ in the optimal solution from (14) (scaled to unit $\ell_2$-norm) maximize/minimize the corresponding $\boldsymbol{\lambda}^T \mathbf{D}_j \mathbf{Xu}_j$ given $(2\mathbf{D}_j - I)\mathbf{Xu}_j^* \ge 0$.

Then, we consider a two-layer ReLU network with $m = 10$ neurons and apply the gradient descent method to train on the logistic loss (3). Let $\hat{\mathbf{w}}_{1,i} = \frac{\mathbf{w}_{1,i}}{\|\mathbf{w}_{1,i}\|_2}$ for $i \in [m]$. We plot $\hat{\mathbf{w}}_{1,i}$ and $(\mathbf{X}\hat{\mathbf{w}}_{1,i})_+$ at iteration $\{10^l | l = 0, \dots, 4\}$ along with neurons in the optimal solution to (14) scaled to unit $\ell_2$-norm in Figure 5. Certain neurons do not move, while the activated neurons trained by gradient descent tend to converge to the direction of the neurons in the optimal solution to (14).

We repeat the training on the logistic loss (3) with the gradient descent method several times and we plot the trajectories in Figure 7.

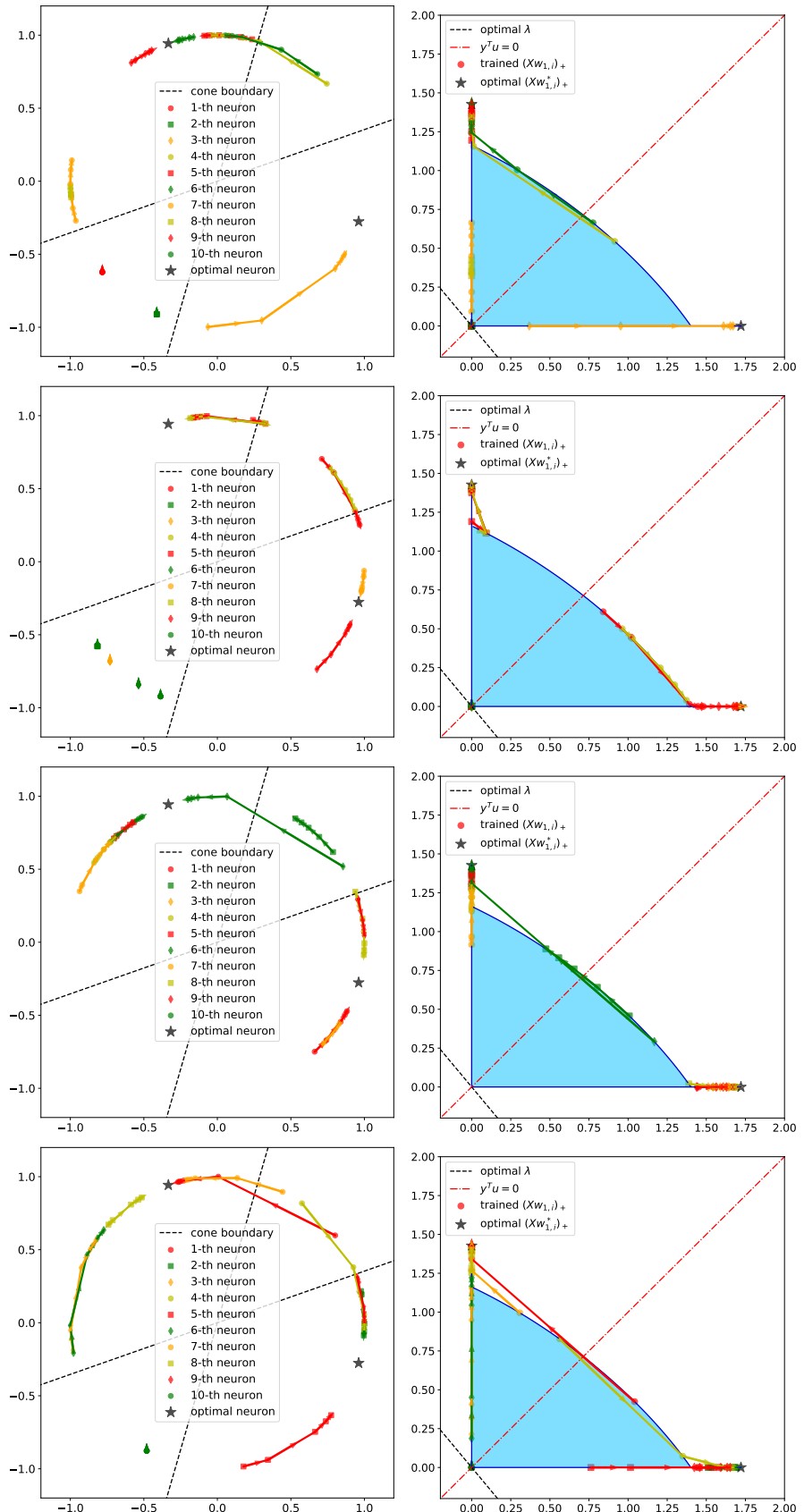

Figure 7: Multiple independent random initializations of gradient descent trajectories on the same orthogonal separable dataset.

## F.2 EXPERIMENT ON SPIKE-FREE DATASET

We repeat the previous numerical experiment on a non-spike-free dataset: $\mathbf{X} = \begin{bmatrix} 1.65 & 0.47 \\ 0.47 & 1.35 \end{bmatrix} \in \mathbb{R}^{2 \times 2}$ and $\mathbf{y} = \begin{bmatrix} 1 \\ 1 \end{bmatrix} \in \mathbb{R}^2$. Similarly, we plot the ellipsoid set and the rectified set in Figure 8.

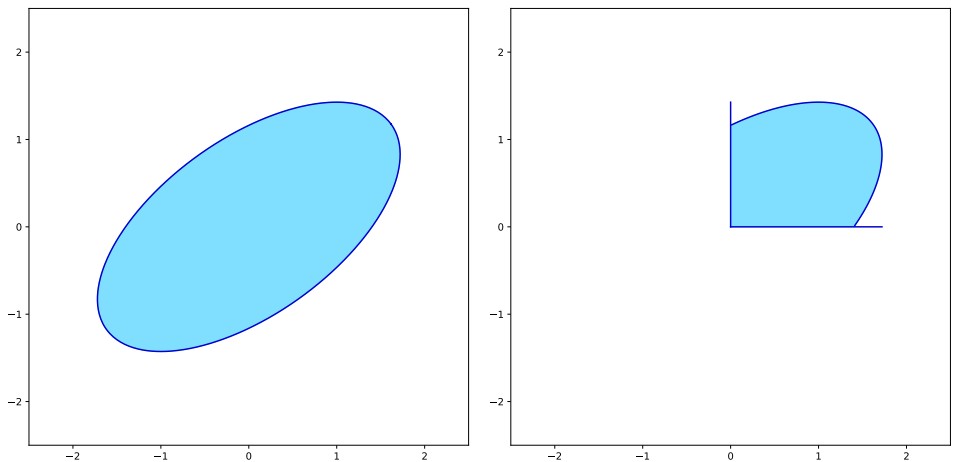

Figure 8: The ellipsoid set and the rectified ellipsoid set for a non-spike-free dataset.

We enumerate all possible hyperplane arrangements in the set $\mathcal{P}$ and solve the convex max-margin problem (14) via CVXPY to obtain the following non-zero neuron

$$\mathbf{u}_{1,4} = \begin{bmatrix} 0.43 \\ 0.59 \end{bmatrix} \tag{130}$$

We plot the rectified ellipsoid set $\{(\mathbf{X}\mathbf{u})_+ | \|\mathbf{u}\|_2 \leq 1\}$, vectors $\mathbf{u}_j$, neurons in the optimal solution to (14) scaled to unit $\ell_2$-norm and the direction of $\boldsymbol{\lambda}$ in Figure 9. We also plot $\hat{\mathbf{w}}_{1,i}$ and $(\mathbf{X}\hat{\mathbf{w}}_{1,i})_+$ at iteration $\{10^l | l = 0, \dots, 4\}$ along with neurons in the optimal solution to (14) scaled to unit $\ell_2$-norm in Figure 10.

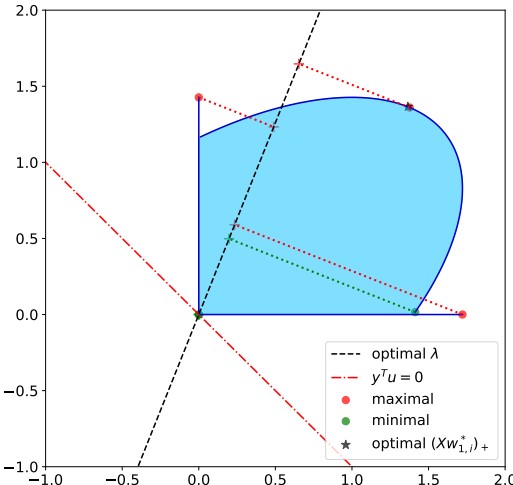

Figure 9: Recitified Ellipsoidal set and corresponding extreme points for a non-spike-free dataset.

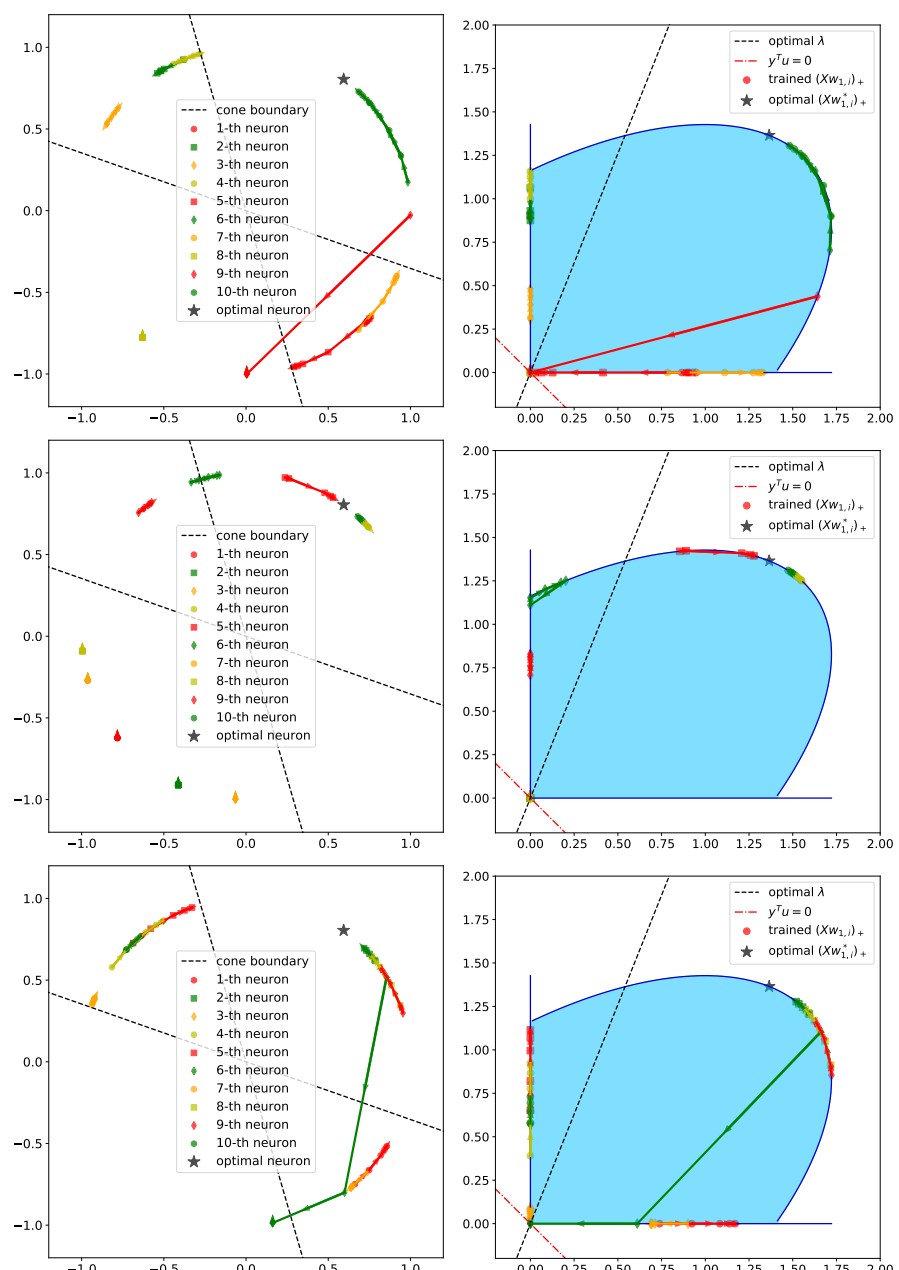

Figure 10: Multiple independent random initializations of gradient descent trajectories on the same non-spike-free dataset. Note that the optimal extreme point (star), which is the uniquely optimal single neuron is on the boundary of the main two-dimensional ellipsoid and not on the one-dimensional spikes (projected ellipsoids). Also note that some neurons are stuck at spurious stationary points.

