# OpenReview forum: "The Convex Geometry of Backpropagation: Neural Network Gradient Flows Converge to Extreme Points of the Dual Convex Program"
_ICLR.cc/2022/Conference — ICLR 2022 Poster_

### Official Review · Reviewer_uQbv · 2021-11-02

**Correctness:** 3
**Technical Novelty And Significance:** 3
**Empirical Novelty And Significance:** Not applicable
**Recommendation:** 6
**Confidence:** 2

**Main Review:**

The main contribution of the paper is to prove that the non-convex gradient flows for the max-margin problem converges to the KKT points of the convex max-margin problem. This is a significant result which may be of interest to parts of the community.

On the other hand, the structure of the paper can be improved. For instance, Section 2.2. is labelled "Outline of our contributions" but this section already contains some preliminary results, e.g. the characterization of the dual extreme points as well as the discussion of the implicit regularization of unregularized gradient flow, while the main contributions are not discussed here.

Moreover, I found Section 5 a bit hard to follow since the purpose of the individual Lemmas is not made clear directly in the section. The flow of the section could be improved by providing additional information about which role they play in the overall proofs.

Finally, the abstract says that the paper presents numerical results verifying the predictions of their theory. However, those experiments are only contained in the Appendix.

Overall, while the paper contains some interesting results, overall the presentation and flow can be improved. Therefore I am leaning towards rejection of the paper.



---
After rebuttal: The authors made several improvements to the flow and structure of the paper, as suggested in my review. Therefore I have increased my score.

**Summary Of The Paper:**

The paper studies the subgradient flows when training a two-layer ReLU neural network. To this end, the non-convex max-margin problem is reformulated as a convex optimization problem. The authors then analyze the dual extreme points of the convex formulation and show the implicit regularization of unregularized gradient flow as convex regularization. Then, for the binary classification problem, it is proven that the KKT points of the non-convex max-margin problem correspond to the KKT points of the convex max-margin problem if the direction is dual feasible. The paper then demonstrates that this is the case under some conditions on spike-free matrices and orthogonal separable data. Finally it is shown that if the dataset is orthogonal separable and initialized sufficiently close to zero, the limiting point of the gradient flow is the global minimizer of the max-margin problem.

**Summary Of The Review:**

The paper contains some interesting results which may be of interest to the community, but the presentation and flow of the paper can be improved.

---

> ### Author Response · Authors · 2021-11-17
> **Response to Reviewer uQbv**
>
> Thanks for your careful reading and constructive comments. We will include the suggestions in the final version. In the remainder, we want to address the main points raised in the reviews.
>
> 1. We made improvements in the structure of the paper as suggested. Earlier, in subsection 2.2, we included a high-level description of the implicit regularization of unregularized gradient flow, and the characterization of the dual extreme points. These are critical to understand the motivation and the proof technique to show that the non-convex subgradient flow globally maximizes the margin. Now, we highlight our main results in the 'Subsection 2.1 our contributions'.
>
> 2. Thanks for your suggestion regarding Section 5. We include short discussions before the technical lemmas to provide intuition and illustrate their purpose.
>
> 3. We present the numerical results in Figure 1, 2, 3 and 4 in the main paper. The detailed description of the experimental setup and and extra numerical results are left to the Appendix. We make sure that the numerical results in the main paper are self-contained.

---

> > ### Comment · Reviewer_uQbv · 2021-11-28
> > **Response to authors**
> >
> > Thank you for responding to my comments and adapting the paper as suggested. I will increase my score.

---

### Official Review · Reviewer_8kmN · 2021-11-04

**Correctness:** 4
**Technical Novelty And Significance:** 4
**Empirical Novelty And Significance:** Not applicable
**Recommendation:** 8
**Confidence:** 3

**Main Review:**

Overall, I feel that the technique and the result are novel and of interest to the community. While other approaches, such as NTK or mean-field, have established global convergence guarantees, those results are asymptotic in the size of the neural network and do not provide the interpretation of the max-margin classifier. I feel that these results make the line of work on the convex formulation more complete, and will serve to further encourage the line of work analyzing neural networks based on some hidden form of convexity.

As an aside, it would be great if the authors can comment on the extendability of this approach to the setup of three-layer ReLU, as studied in Ergen and Pilanci 2021.

**Summary Of The Paper:**

This work analyzes the training dynamics of two-layer ReLU networks applied to separable problem data, based on the equivalent convex formulation by Pilanci, Ergen 2020. The main result states that the gradient-flow training dynamics provably converges to a "maximum-margin classifier".

**Summary Of The Review:**

Establishing training guarantees of two-layer ReLU networks based on convex formulation of Pilanci, Ergen 2020 is sufficiently novel and interesting.

---

> ### Author Response · Authors · 2021-11-17
> **Response to Reviewer 8kmN**
>
> Thanks for your careful reading and constructive comments. We will include the suggestions in the final version. In the remainder, we want to address the main points raised in the reviews.
>
> It is challenging but possible to extend our results to three-layer and deeper neural networks. Some of our results leverage the analysis in [1] and [2] for two-layer networks, where the subgradient of the non-convex loss with respect to each neuron is related to the dual variable of the convex max-margin problem. It requires extra work to extend these results to deeper networks.
>
> [1] Kaifeng Lyu and Jian Li. Gradient descent maximizes the margin of homogeneous neural networks.
>
> [2] Mary Phuong and Christoph H Lampert.  The inductive bias of relu networks on orthogonallyseparable data.

---

### Official Review · Reviewer_pBSm · 2021-11-08

**Correctness:** 4
**Technical Novelty And Significance:** 3
**Empirical Novelty And Significance:** 3
**Recommendation:** 6
**Confidence:** 3

**Main Review:**

Strengths:
1. Clear presentation and strong logicality;
2. The corresponding non-convex max-margin problem with the following dual reformulation is clear (maybe mathematically correct);
3. The geometry discussion of the neural gradient flow is clear, especially based on Figures 1-4.


**Summary Of The Paper:**

This paper discussed the convergence of the gradient flows of two-layer neural networks, while they claim the convergence of non-convex gradient flows to global optimum which is different from existing works. The theoretical analysis framework seems to be novel and some sufficient condition is proposed to support the strong theoretical results.

**Summary Of The Review:**

Although I am not an expert on this topic, the paper is well-written and the novelty of the idea seems to be clear after careful reading and thinking.

---

> ### Author Response · Authors · 2021-11-17
> **Response to Reviewer pBSm**
>
> Thanks for your careful reading and constructive comments.

---

### Decision · Program_Chairs · 2022-01-20

**Decision:**

Accept (Poster)

**Comment:**

The papers makes progress on the important question of implicit bias in gradient based neural learning. Remarkably they derive reasonable conditions for global optimality.